# Maximal Correlation-Based Post-Nonlinear Learning for Bivariate Causal Discovery

## Abstract

Bivariate causal discovery aims to determine the causal relationship between two random variables from passive observational data (as intervention is not affordable in many scientific fields), which is considered fundamental and challenging. Designing algorithms based on the post-nonlinear (PNL) model has aroused much attention for its generality. However, the state-of-the-art (SOTA) PNL-based algorithms involve highly non-convex objectives due to the use of neural networks and non-convex losses, thus optimizing such objectives is often time-consuming and unable to produce meaningful solutions with finite samples. In this paper, we propose a novel method that incorporates maximal correlation into the PNL model learning (short as MC-PNL) such that the underlying nonlinearities can be accurately recovered. Owing to the benign structure of our objective function, when modeling the nonlinearities with linear combinations of random Fourier features, the target optimization problem can be solved rather efficiently and rapidly via the block coordinate descent. We also compare the MC-PNL with SOTA methods on the downstream synthetic and real causal discovery tasks to show its superiority in time and accuracy. Our code is available at https://anonymous.4open.science/r/MC-PNL-E446/ .

## 1 Introduction and Related Works

Causal discovery is an old and new topic to the machine learning community, which aims to find causal relationships among variables. Many recent attempts at application have emerged in various scientific domains, such as climate science (Ebert-Uphoff & Deng, 2012; Runge et al., 2019), bioinformatics (Choi et al., 2020; Foraita et al., 2020; Shen et al., 2020), etc. The gold standard for causal discovery is to conduct randomized experiments (via interventions), however, interventions are often expensive, unethical, and impractical. It is highly demanded to discover causal relationships purely from passive observational data. In the past three decades, many pioneer algorithms for directed acyclic graph (DAG) searching have been developed for multi-variate causal discovery to reduce the computational complexity and improve the accuracy. For example, there are constraint/independence-based algorithms such as IC, PC, FCI (Pearl, 2009; Spirtes et al., 2000), RFCI (Colombo et al., 2012) (too many to be listed), as well as score-based methods such as GES (Chickering, 2002), NOTEARS (Zheng et al., 2018), etc. However, the algorithms mentioned above can merely return a Markov equivalence class (MEC) that encodes the same set of conditional independencies, with many undetermined edge directions; moreover, the discovered DAG may not necessarily be causal. In this paper, we will focus on a fundamental problem, namely bivariate causal discovery, which aims to determine the causal direction between two random variables $X$ and $Y$. Bivariate causal discovery is one promising routine for further identification of the underlying *causal* DAG (Peters et al., 2017).

Bivariate causal discovery is a challenging task, which cannot be directly solved using the existing methodologies for the multivariate case, as the two candidate DAGs, $X \to Y$ and $X \leftarrow Y$, are in the same MEC. More assumptions should be imposed to make bivariate causal discovery feasible, as summarized by Peters et al. (2017). One assumption is on the *a priori* model class restriction, e.g., linear non-Gaussian acyclic model (LiNGAM) (Shimizu et al., 2006), nonlinear additive noise model (ANM) (Mooij et al., 2016), post-nonlinear (PNL) model (Zhang & Hyvärinen, 2009), etc. The other assumption is on the "independence of cause and mechanism" leading to the algorithms of trace condition (Janzing et al., 2010), IGCI (Janzing et al., 2012), distance correlations (Liu & Chan, 2016), meta-transfer (Bengio et al., 2020), CDCI (Duong & Nguyen, 2022), etc. There are

also seminal works focusing on causal discovery in linear/nonlinear dynamic systems, which are out of the scope of this paper, and the corresponding representatives are Granger causality test (Granger, 1969) and convergent cross mapping (Sugihara et al., 2012; Ye et al., 2015).

In this work, we focus on the PNL model, which is more general than LiNGAM and ANM. The existing works merely show the identifiability results with infinite data samples (i.e. known joint distribution), while practical issues with finite sample size are seldom discussed. We reveal the difficulties with the current PNL-based algorithms in the finite sample regime, such as insufficient model fitting, slow training progress, and unsatisfactory independent test performance, and correspondingly propose novel and practical solutions.

The main contributions of this work are as follows.

1. We point out various practical training issues with the existing PNL model learning algorithms, in particular PNL-MLP and AbPNL, and propose a new algorithm called MC-PNL (specifically the **maximal correlation**-based algorithm with **independence regularization**), which can achieve a better recovery of the underlying nonlinear transformations.

2. We suggest using the randomized dependence coefficient (RDC) instead of the Hilbert-Schmidt independence criterion (HSIC) for the independent test and give a universal view of some widely used dependence measures.

3. We use MC-PNL for model learning in bivariate causal discovery and show that our method outperforms other SOTA independence test-based methods on various benchmark datasets.

## 2 PRELIMINARIES

In this section, we will introduce the HSIC as a dependence measure, the current HSIC-based causal discovery methods for PNL model, and other relevant learning methods based on the Hirschfeld-Gebelein-Rényi (HGR) correlation. Our proposed MC-PNL method exploits all these ingredients.

### 2.1 HSIC SCORE AND HSIC-BASED REGRESSION

Regression by dependence minimization (Mooij et al., 2009) has attracted lots of attention recently. Greenfeld & Shalit (2020) has shown its power for robust learning, in particular the unsupervised covariate shift task. Let us consider the following regression model,

$$Y = f(X) + \epsilon, \quad \epsilon \perp\!\!\!\perp X, \tag{1}$$

where the additive noise $\epsilon$ is independent (symbolized by $\perp\!\!\!\perp$) with the input variable $X$, and the selected regression model $f_{\boldsymbol{\theta}}$ is to be learned via **minimizing the dependency** between the input variable $X$ and the residual $Y - f_{\boldsymbol{\theta}}(X)$. A widely used dependence measure is the Hilbert-Schmidt independence criterion (HSIC) (Gretton et al., 2005; 2007).

**Definition 1** (HSIC). *Let $X, Z \sim P_{XZ}$ be jointly distributed random variables, and $\mathcal{F}, \mathcal{G}$ be the reproduced kernel Hilbert spaces with kernel functions $k$ and $l$, the HSIC can be expressed as,*

$$\begin{aligned} \mathrm{HSIC}(X, Z; \mathcal{F}, \mathcal{G}) =& \mathbb{E}_{XZ}\mathbb{E}_{X'Z'}k\left(x, x'\right)l\left(z, z'\right) + \mathbb{E}_X\mathbb{E}_{X'}k\left(x, x'\right)\mathbb{E}_Z\mathbb{E}_{Z'}l\left(z, z'\right) \\ &- 2\mathbb{E}_{X'Z'}\left[\mathbb{E}_X k\left(x, x'\right)\mathbb{E}_Z l\left(z, z'\right)\right], \end{aligned} \tag{2}$$

*where $x'$ and $z'$ denote independent copies of $x$ and $z$, respectively.*

**Remark 2.1.** *We can conclude that: (a) $X \perp\!\!\!\perp Z \Rightarrow \mathrm{HSIC}(X, Z) = 0$; (b) with a proper universal kernel (e.g., Gaussian kernel), $X \perp\!\!\!\perp Z \Leftarrow \mathrm{HSIC}(X, Z) = 0$ (Gretton et al., 2005).*

When the joint distribution $P_{XZ}$ is unknown, given a dataset with $n$ samples ($\mathbf{x} = [x_1, x_2, \ldots, x_n]^T \in \mathbb{R}^n$, $\mathbf{z} = [z_1, z_2, \ldots, z_n]^T \in \mathbb{R}^n$), a biased HSIC estimate can be constructed as,

$$\widehat{\mathrm{HSIC}}\left(\mathbf{x}, \mathbf{z}; \mathcal{F}, \mathcal{G}\right) = \frac{1}{n^2}\operatorname{tr}(KHLH), \tag{3}$$

where $K_{i,j} = k\left(x_i, x_j\right)$, $L_{i,j} = l\left(z_i, z_j\right)$, and $H = I - \frac{1}{n}\mathbf{1}\mathbf{1}^T \in \mathbb{R}^{n \times n}$ is a centering matrix. The Gaussian kernel $k\left(x_i, x_j\right) = \exp\left(-(x_i - x_j)^2\sigma^{-2}\right)$ is commonly used, and the same for $l$. One

can intuitively interpret this empirical HSIC as the inner-product of two centralized kernel matrices $\frac{1}{n^2}\langle HKH, HLH\rangle$, where the kernel matrices summarize the sample similarities.

Mooij et al. (2009) first proposed to use the above defined empirical HSIC for model learning. Concretely, the regression model is a linear combination of the basis functions, $f_{\boldsymbol{\theta}}(x) = \sum_{i=1}^{k} \theta_i \phi_i(x)$, and the parameters are learned from:

$$\hat{\boldsymbol{\theta}} \in \arg\min_{\boldsymbol{\theta}\in\mathbb{R}^p} \left( \widehat{\mathrm{HSIC}}(\mathbf{x}, \mathbf{y} - f_{\boldsymbol{\theta}}(\mathbf{x})) + \frac{\lambda}{2}\|\boldsymbol{\theta}\|_2^2 \right), \tag{4}$$

where $f_{\boldsymbol{\theta}}$ is applied elementwisely to the data points, and $\lambda > 0$ is a penalty parameter (we will keep using $\lambda$ as a penalty parameter under different contexts). One key advantage of this formulation is that it requires no assumption on the noise distribution. Greenfeld & Shalit (2020) implemented $f_{\boldsymbol{\theta}}$ using neural networks, and showed the learnability of the HSIC loss theoretically.

## 2.2 Causal Discovery with Post-Nonlinear Model

The bi-variate post-nonlinear model is expressed as, $Y = f_2(f_1(X) + \epsilon)$, where $f_1$ denotes the nonlinear effect of the cause, $\epsilon$ is the independent noise, and $f_2$ denotes the invertible post-nonlinear distortion from the sensor or measurement side. The goal is to find the causal direction $X \to Y$ from a set of passive observations on $X$ and $Y$. Note that from the data generating process, $\epsilon$ is independent with $X$ but not $Y$. Taking this asymmetry as a prior, one can test the causal direction by first learning the underlying transformations, $f_2^{-1}$ and $f_1$, and then checking the independence between the residual $r_{(\to)} = f_2^{-1}(Y) - f_1(X)$ and the input $X$.

The PNL-MLP algorithm proposed by Zhang & Hyvärinen (2009) tests between two hypotheses ($X \to Y$ and $X \leftarrow Y$) as follows. Under the hypothesis $X \to Y$, one can parameterize $f_1$ and $f_2^{-1}$ by two multi-layer perceptrons (MLPs) $f_{(\to)}$ and $g_{(\to)}$, and learn them via minimizing the mutual information (MI):

$$\hat{f}_{(\to)}, \hat{g}_{(\to)} \in \arg\min_{f_{(\to)}, g_{(\to)}} \mathrm{MI}\left(\mathbf{r}_{(\to)} := g_{(\to)}(\mathbf{y}) - f_{(\to)}(\mathbf{x}); \mathbf{x}\right), \tag{5}$$

where $g_{(\to)}, f_{(\to)}$ are applied elementwisely. The estimated residual is $\hat{\mathbf{r}}_{(\to)} = \hat{g}_{(\to)}(\mathbf{y}) - \hat{f}_{(\to)}(\mathbf{x})$. Similarly, under the hypothesis $X \leftarrow Y$, one can obtain an estimate of $\hat{\mathbf{r}}_{(\leftarrow)} = \hat{g}_{(\leftarrow)}(\mathbf{x}) - \hat{f}_{(\leftarrow)}(\mathbf{y})$ via minimizing $\mathrm{MI}(\mathbf{r}_{(\leftarrow)}; \mathbf{y})$. The causal direction is determined by comparing $\widehat{\mathrm{HSIC}}\left(\hat{\mathbf{r}}_{(\to)}, \mathbf{x}\right)$ and $\widehat{\mathrm{HSIC}}\left(\hat{\mathbf{r}}_{(\leftarrow)}, \mathbf{y}\right)$. If $\widehat{\mathrm{HSIC}}\left(\hat{\mathbf{r}}_{(\to)}, \mathbf{x}\right) < \widehat{\mathrm{HSIC}}\left(\hat{\mathbf{r}}_{(\leftarrow)}, \mathbf{y}\right)$, the hypothesis $X \to Y$ is endorsed; otherwise, the hypothesis $X \leftarrow Y$ is endorsed.

However, the MI between random variables is often difficult to calculate (see supplement A), and tuning the MLPs requires many tricks as mentioned in Zhang & Hyvärinen (2009), altogether bringing huge difficulties to handle large-scale datasets with many variable pairs. Uemura & Shimizu (2020) proposed AbPNL method that uses HSIC instead of MI, and imposes the invertibility restriction of $f_2$ via an auto-encoder to eliminate nonsense solutions,

$$\min_{f,g,g'} \widehat{\mathrm{HSIC}}\left(\mathbf{x}, \mathbf{r} := g(\mathbf{y}) - f(\mathbf{x})\right) + \lambda\|\mathbf{y} - g'(g(\mathbf{y}))\|_2^2, \tag{6}$$

where $g, g'$ are encoder and decoder MLPs. The subscript $_{(\to)}$ is omitted for conciseness here.

We summarize the architectures of the above-mentioned two methods in Figure 1. Nevertheless, inherent issues exist concerning the cost function and the neural network training procedure when dealing with finite sample datasets, see in Section 3.1.

## 2.3 PNL Learning through Maximal Correlation

Another routine to learn the nonlinear transformations $f$ and $g$ is through the HGR maximal correlation (Hirschfeld, 1935; Gebelein, 1941; Rényi, 1959).

**Definition 2** (HGR maximal correlation). *Let $X, Y$ be jointly distributed random variables. Then,*

$$\rho^* = \text{HGR}(X;Y) := \sup_{\substack{f:\mathcal{X}\to\mathbb{R},g:\mathcal{Y}\to\mathbb{R} \\ \mathbb{E}[f(X)]=\mathbb{E}[g(Y)]=0 \\ \mathbb{E}[f^2(X)]=\mathbb{E}[g^2(Y)]=1}} \mathbb{E}[f(X)g(Y)], \tag{7}$$

*is the HGR maximal correlation between $X$ and $Y$, and $f,g$ are the associated maximal correlation functions.*

**Remark 2.2.** *The HGR maximal correlation $\rho^*$ is attractive as a measure of dependency due to some useful properties: (1) Bounded $\rho^*$ : $0 \le \rho^* \le 1$; (2) $X$ and $Y$ are independent if and only if $\rho^* = 0$; (3) there exists $f$ and $g$ such that $f(X) = g(Y)$ with probability 1 if and only if $\rho^* = 1$.*

The optimal unit-variance feature transformations, $f^*$ and $g^*$, can be found by iteratively updating $f$ and $g$ in (7). However, for causal discovery applications, one fatal issue is that the learned $f^*$ and $g^*$ are constrained to have unit-variance, thus being unable to reflect the true magnitudes of the underlying functions $f$ and $g$. As a consequence, the resulting residual can be incorrect for the independence tests in the next stage. We found two possible remedies in the literature, namely the alternating conditional expectation (ACE) algorithm (Breiman & Friedman, 1985) and a soft version of (7) (Soft-HGR)(Wang et al., 2019).

The ACE algorithm solves the regression problem (8) by computing the conditional mean alternatively,

$$\min_{f,g} \mathbb{E}(f(X) - g(Y))^2, \quad \text{s.t.} \quad \mathbb{E}[f(X)] = \mathbb{E}[g(Y)] = 0, \quad \mathbb{E}[g^2(Y)] = 1, \tag{8}$$

which only retains the unit-variance constraint on $g$. The equivalence to (7) was established, and the regression optimal transformation $(f^{**}, g^{**})$ equals $(\rho^* f^*, g^*)$, see Theorem 5.1 in Breiman & Friedman (1985). The other formulation, Soft-HGR, relaxes the unit-variance constraints as follows,

$$\max_{f,g} \mathbb{E}\left[f(X)g(Y)\right] - \frac{1}{2}\,\text{var}(f(X))\,\text{var}(g(Y)), \quad \text{s.t.} \quad \mathbb{E}[f(X)] = \mathbb{E}[g(Y)] = 0. \tag{9}$$

It allows certain linear transformations $(af^*, a^{-1}g^*)$, where $a \in \mathbb{R}\backslash\{0\}$ can produce infinitely many equivalent local minima. This scale ambiguity results in enormous useless solutions for causal discovery, and the desired one should make the estimated residual independent with the input. We will show how our proposed method is able to eliminate those undesired solutions in Section 4.

**Connections to VICReg.** We notice that the recent proposed Variance-Invariance-Covariance Regularization (VICReg) (Bardes et al., 2022) shares similar intuitions with the HGR maximal correlation. When the dimension of representation vectors (i.e., $f$ and $g$) reduces to one, the covariance term disappears, and the VICReg objective becomes,

$$\min_{f,g} \underbrace{\mathbb{E}(f(X) - g(Y))^2}_{\text{invariance term}} + \lambda \underbrace{\left[\max(0, \gamma - \text{var}(f(X))) + \max(0, \gamma - \text{var}(g(Y)))\right]}_{\text{variance term}}, \tag{10}$$

where $\lambda, \gamma > 0$ are the hyper-parameters that need to be tuned. The invariance term encourages the alignment of the learned features; and the variance term encourages a $\gamma$-bounded variation to avoid trivial solutions like $f(X) = g(Y) = $ constant. To see the connections, we rewrite Soft-HGR (9) as,

$$\min_{f,g} \quad \underbrace{\mathbb{E}\left[f(X) - g(Y)\right]^2}_{\text{invariance term}} + \underbrace{\text{var}(f(X))\,\text{var}(g(Y)) - \text{var}(f(X)) - \text{var}(g(Y))}_{\text{variance term}},$$
$$\text{s.t.} \quad \mathbb{E}[f(X)] = \mathbb{E}[g(Y)] = 0, \tag{11}$$

in which the variance of $f$ and $g$ are also encouraged but not allowed to grow simultaneously.

## 3 PRACTICAL ISSUES WITH EXISTING ALGORITHMS

In this section, we summarize several practical issues of the existing algorithms for PNL learning, including among others PNL-MLP (Zhang & Hyvärinen, 2010) and AbPNL (Uemura & Shimizu, 2020). These issues motivate our novel MC-PNL method to be introduced in Section 4, see the comparisons in terms of their architectures in Figure 1.

### 3.1 ISSUES ON MODEL LEARNING

**Over-fitting issue.** The general idea of the PNL model learning, according to Section 2.2, is to encourage the independence between the input and the estimated residual. Both PNL-MLP and AbPNL use neural networks to parameterize $f$ and $g$. But it is skeptical that meaningful representations can really be learned with finite samples. Let us review the dependence minimization problem below,

$$\min_{f,g} \widehat{\text{HSIC}}(\mathbf{x}, \mathbf{r}) = \frac{1}{n^2} \text{tr}(K_{\mathbf{xx}} H L_{\mathbf{rr}} H), \quad \text{where } \mathbf{r} = f(\mathbf{x}) - g(\mathbf{y}). \tag{12}$$

We argue that it is utmost difficult to learn meaningful representations of $f$ and $g$ via minimizing solely the HSIC score, due to the enormous degrees of freedom for $f$ and $g$ to learn arbitrary random noise. We conducted experiments using both wide over-parameterized and narrow deep neural networks with sufficient representation power. In our simulation results (see supplement B), for both network architectures, the objective values can reach zero but unfortunately produce meaningless estimates. This is unsurprising though, as one can force $\mathbf{r}$ to be samples from arbitrary independent random noise (Yun et al., 2019; Zhang et al., 2021). To aid with that, we propose to cooperate dependence minimization with maximal correlation, which helps to obtain desired solutions, see Figure 1(c) for illustration and Section 4 for details.

**Optimization issue.** The optimization of neural networks is a long-standing problem, and yet there is not any study on the optimization landscape of the HSIC loss with neural networks. Typically, first-order methods such as stochastic gradient descent are used in the existing causal discovery methods, and initialization is crucial to the causal discovery accuracy, see in supplement B. In this paper, we suggest parameterizing both $f$ and $g$ as a linear combination of random Fourier features and using a linear kernel for HSIC, which admit a benign landscape with symmetry (see Chapter 7 in Wright & Ma (2022)) for the non-convex optimization.

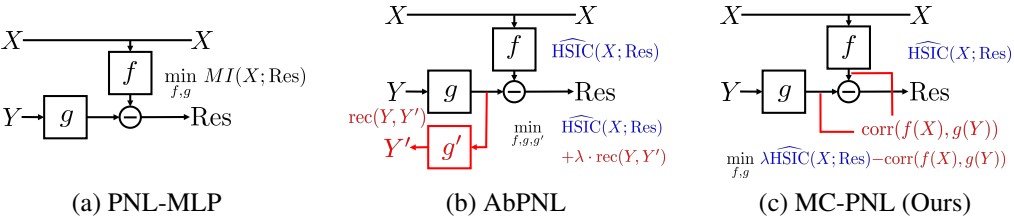

Figure 1: Architectures of PNL learning frameworks.

### 3.2 ISSUES ON INDEPENDENCE TEST

As the independence test is critical to the accuracy of causal discovery, we have to cautiously choose the dependence measure. Although HSIC is widely used, there are several drawbacks of HSIC (e.g., the choice of kernel and corresponding hyper-parameters are user-defined, the values of HSIC depends on the scale of the random variables). In this section, we show experimentally that the HSIC score is not the best choice, and we favor randomized dependence coefficient (RDC) (Lopez-Paz et al., 2013) particularly for finite samples.

We generated various synthetic datasets following the PNL models, see supplement C, in which we know in advance that the injected noise $\epsilon \perp\!\!\!\perp X$ and $\epsilon \not\!\perp\!\!\!\perp Y$. Thus, we are able to compare various dependence measures, by checking whether $\text{Dep}(\mathbf{x}, \boldsymbol{\epsilon}) < \text{Dep}(\mathbf{y}, \boldsymbol{\epsilon})$ on various datasets. In this section, the compared dependence measures are HSIC (Gretton et al., 2005), its normalized variant (NOCCO) (Fukumizu et al., 2007), and RDC (Lopez-Paz et al., 2013). Besides, we also study the impact of different choices of linear, Gaussian radial basis function (RBF), and rational quadratic (RQ) kernels. We note here that RDC is a computational tractable estimator inspired by the HGR maximal correlation. It shows that RDC outperforms other dependence measures especially when the sample size is small, see Table 1. **Thus, we advocate to use RDC to measure dependency with finite samples.** Finally, we give a universal view of the aforementioned dependence measures in supplement D.

Table 1: The independence test accuracy (%) with known injected noise

| # samples | $n = 1000$ | | | $n = 2000$ | | | $n = 5000$ | | |
|---|---|---|---|---|---|---|---|---|---|
| noise level $\sigma_\epsilon$ | 0.1 | 1 | 10 | 0.1 | 1 | 10 | 0.1 | 1 | 10 |
| HSIC-linear | 79 | 95 | 100 | 88 | 96 | 100 | 85 | 97 | 100 |
| HSIC-RBF | 78 | 97 | 100 | 85 | 99 | 100 | 86 | 97 | 100 |
| HSIC-RQ | 79 | 98 | 100 | 86 | 99 | 100 | 91 | 96 | 100 |
| NOCCO-RBF | 87 | 83 | 75 | 83 | 85 | 72 | 90 | 92 | 90 |
| NOCCO-RQ | 87 | 82 | 73 | 81 | 86 | 65 | 85 | 94 | 87 |
| **RDC** | **94** | **100** | **100** | **98** | **100** | **100** | **100** | **100** | **100** |

## 4 PROPOSED METHOD

In this section, we propose a maximal correlation-based post-nonlinear model learning framework, called MC-PNL, to accurately estimate the nonlinear functions and compute the corresponding residuals. After then, independence tests will be conducted to determine the causal direction.

### 4.1 MAXIMAL CORRELATION-BASED PNL MODEL LEARNING

As we can see in the previous sections, minimizing HSIC (12) requires no assumption on the noise distribution and encourages the independence of the residual, but it can easily get stuck at meaningless local minima. Maximal HGR correlation based methods can learn meaningful transformations as its name suggested, but not necessarily produce independent residual.

To combine their strengths, we propose the following MC-PNL objective,

$$\min_{f,g} \quad -\mathbb{E}\left[f(X)g(Y)\right] + \tfrac{1}{2}\operatorname{var}(f(X))\operatorname{var}(g(Y)) + \lambda\operatorname{Dep}(X, f(X) - g(Y)),$$
$$\text{s.t.} \quad \mathbb{E}[f(X)] = \mathbb{E}[g(Y)] = 0, \tag{13}$$

where $\operatorname{Dep}(\cdot, \cdot) \geq 0$ is a dependence measure (e.g., HSIC with different kernel functions), and $\lambda > 0$ is a hyper-parameter that penalizes the dependence between the input variable $X$ and the estimated residuals $f(X) - g(Y)$. This novel objective can learn meaningful feature transformations with the Soft-HGR term, and resolve the scale ambiguity via the dependence minimization term. The objective (13) is consistent with minimizing MI principle, under the assumptions of invertible PNL generating functions and Gaussian noise, see details in supplement E.

**Parameterization with Random Features**

For ease of optimization, we parameterize the transformation functions as the linear combination of the random features, namely $f(x; \boldsymbol{\alpha}) := \boldsymbol{\alpha}^T \boldsymbol{\phi}(x)$ and $g(y; \boldsymbol{\beta}) := \boldsymbol{\beta}^T \boldsymbol{\psi}(y)$ , where the random features $\boldsymbol{\phi}(x) \in \mathbb{R}^{k_1}, \boldsymbol{\psi}(y) \in \mathbb{R}^{k_2}$ are nonlinear projections as described in López-Paz et al. (2013), see supplement F. For a given dataset $\{(x_i, y_i)\}_{i=1}^n$, the corresponding feature matrices are denoted as $\Phi := [\boldsymbol{\phi}(x_1), \boldsymbol{\phi}(x_2), \ldots, \boldsymbol{\phi}(x_n)] \in \mathbb{R}^{k_1 \times n}$ and $\Psi := [\boldsymbol{\psi}(y_1), \boldsymbol{\psi}(y_2), \ldots, \boldsymbol{\psi}(y_n)] \in \mathbb{R}^{k_2 \times n}$. We further denote the residual vector as $\mathbf{r} := \Phi^T \boldsymbol{\alpha} - \Psi^T \boldsymbol{\beta}$.

Consequently, (13) can be written as the following non-convex programming problem,

$$\min_{\boldsymbol{\alpha}, \boldsymbol{\beta}} \quad J(\boldsymbol{\alpha}, \boldsymbol{\beta}) := -\tfrac{1}{n}\boldsymbol{\alpha}^T \Phi \Psi^T \boldsymbol{\beta} + \tfrac{1}{2n^2}\boldsymbol{\alpha}^T \Phi \Phi^T \boldsymbol{\alpha} \boldsymbol{\beta}^T \Psi \Psi^T \boldsymbol{\beta} + \lambda \operatorname{Dep}(\mathbf{x}, \mathbf{r})$$
$$\text{s.t.} \quad \boldsymbol{\alpha}^T \Phi \mathbf{1} = \boldsymbol{\beta}^T \Psi \mathbf{1} = 0, \tag{14}$$

where $\mathbf{1}$ is an all-ones vector, and the dependence measure $\operatorname{Dep}(\mathbf{x}, \mathbf{r})$ can be specially set to the HSIC score with linear kernel, namely,

$$
\begin{aligned}
\widehat{\operatorname{HSIC}}^{lin}(\mathbf{x}, \mathbf{r}) &= \frac{1}{n^2}\operatorname{tr}(K_{\mathbf{xx}} H L_{\mathbf{rr}}^{lin} H) = \frac{1}{n^2}\operatorname{tr}(K_{\mathbf{xx}} H \mathbf{r}\mathbf{r}^T H) \\
&= \frac{1}{n^2}\operatorname{tr}(K_{\mathbf{xx}} H (\Phi^T \boldsymbol{\alpha} - \Psi^T \boldsymbol{\beta})(\Phi^T \boldsymbol{\alpha} - \Psi^T \boldsymbol{\beta})^T H) \\
&= \frac{(\boldsymbol{\alpha}^T \Phi H K_{\mathbf{xx}} H \Phi^T \boldsymbol{\alpha} + \boldsymbol{\beta}^T \Psi H K_{\mathbf{xx}} H \Psi^T \boldsymbol{\beta} - 2\boldsymbol{\alpha}^T \Phi H K_{\mathbf{xx}} H \Psi^T \boldsymbol{\beta})}{n^2}.
\end{aligned} \tag{15}
$$

**Remark:** We adopt the HSIC with linear kernel $L_{\mathbf{rr}}^{lin}$ mainly for a favorable optimization structure, as the resulting HSIC score admits a quadratic form w.r.t. both $\boldsymbol{\alpha}$ and $\boldsymbol{\beta}$. Note that the penalty HSIC term is always non-negative, but the Soft-HGR objective can be negative.

The above problem can be solved via a simple block coordinate descent (BCD) algorithm that updates $\boldsymbol{\alpha}$ and $\boldsymbol{\beta}$ iteratively, see Algorithm 1. Essentially, (14) is multi-convex (Xu & Yin, 2013), and in each update (line 3 or 4 in Algorithm 1), the sub-problem is a linearly constrained quadratic programming. When the sub-problem is strictly convex, one can obtain the unique minimum in closed-form in each update, which admits convergence guarantee to a critical point (Grippo & Sciandrone, 2000). More details on the subproblem optimization and the landscape study can be found in supplement G.

---

**Algorithm 1** BCD for problem 14

---

1: Initialize $\boldsymbol{\alpha}^{(0)}$ and $\boldsymbol{\beta}^{(0)}$ // *Use random initialization*
2: **for** t=1:T **do**
3:     Update $\boldsymbol{\alpha}^{(t)} \leftarrow \arg\min_{\boldsymbol{\alpha}} J(\boldsymbol{\alpha}, \boldsymbol{\beta}^{(t-1)})$,   s.t.   $\boldsymbol{\alpha}^T \Phi \mathbf{1} = 0.$
4:     Update $\boldsymbol{\beta}^{(t)} \leftarrow \arg\min_{\boldsymbol{\beta}} J(\boldsymbol{\alpha}^{(t)}, \boldsymbol{\beta})$,   s.t.   $\boldsymbol{\beta}^T \Psi \mathbf{1} = 0.$
5:     **if** stopping creteria is met **then**
6:         **return** $\boldsymbol{\alpha}^{(t)}, \boldsymbol{\beta}^{(t)}$
7:     **end if**
8: **end for**

---

**Remark:** We can also impose the invertability of $g$ by limiting the derivative $\frac{d}{dy}g(y)$ to be positive (or negative) in line 4, i.e., $\tilde{\Psi}^T \boldsymbol{\beta} > \mathbf{0}$, where $\tilde{\Psi} = [\frac{d}{dy}\boldsymbol{\psi}(y_1), \frac{d}{dy}\boldsymbol{\psi}(y_2), \ldots, \frac{d}{dy}\boldsymbol{\psi}(y_n)] \in \mathbb{R}^{k_2 \times n}$.

**Fine-tune:** Algorithm 1 may produce solutions with distortions, see Figure 2, probably due to the use of the linear kernel. To cope with that, one can enlarge the penalty of dependence $\lambda$, and use HSIC with universal kernels or other dependence measures. Besides, we propose a banded loss to reinforce a banded residual plot, see in supplement H.

## 4.2 DISTINGUISH CAUSE FROM EFFECT VIA INDEPENDENCE TEST

Following the framework proposed by Zhang & Hyvärinen (2009), we distinguish cause from effect according to Algorithm 2. We first fit nonlinear models $f_{(\to)}, g_{(\to)}$ under hypothesis $X \to Y$, and $f_{(\leftarrow)}, g_{(\leftarrow)}$ under hypothesis $X \leftarrow Y$. After the learning iterations, we conduct independence tests. If $\widehat{\text{Dep}}\left(\hat{\mathbf{r}}_{(\to)}, \mathbf{x}\right) < \widehat{\text{Dep}}\left(\hat{\mathbf{r}}_{(\leftarrow)}, \mathbf{y}\right)$, the hypothesis $X \to Y$ is supported; otherwise, the hypothesis $X \leftarrow Y$ is supported. We use the RDC for the independent test, as introduced in Section 3.2.

---

**Algorithm 2** The MC-PNL method for causal direction prediction.

---

**Input:** The standardized data $\mathbf{x}, \mathbf{y} \in \mathbb{R}^n$.
**Output:** The causal score $C_{X \to Y}$ and direction.
  1. Fit PNL models via Algorithm 1 and estimate residuals under hypotheses, $X \to Y$ and $X \leftarrow Y$.
     • Under hypothesis $X \to Y$: $\hat{\mathbf{r}}_{(\to)} = \hat{g}_{(\to)}(\mathbf{y}) - \hat{f}_{(\to)}(\mathbf{x})$.
     • Under hypothesis $X \leftarrow Y$: $\hat{\mathbf{r}}_{(\leftarrow)} = \hat{g}_{(\leftarrow)}(\mathbf{x}) - \hat{f}_{(\leftarrow)}(\mathbf{y})$.
  2. Calculate the causal score $C_{X \to Y} := \widehat{\text{Dep}}\left(\hat{\mathbf{r}}_{(\leftarrow)}, \mathbf{y}\right) - \widehat{\text{Dep}}\left(\hat{\mathbf{r}}_{(\to)}, \mathbf{x}\right).$
  3. Output the causal score $C_{X \to Y}$ and

$$\text{direction} := \begin{cases} X \to Y, & \text{if } C_{X \to Y} > 0, \\ X \leftarrow Y, & \text{if } C_{X \to Y} < 0, \end{cases}$$

---

Towards trustworthy decisions, Liu & Chan (2016) proposed to make no decision when $|C_{X \to Y}|$ is less than a threshold $\delta > 0$. Besides, bootstrap (Efron, 1992; Zoubir & Boashash, 1998) can also be used for uncertainty quantification, see supplement I.

## 5 EXPERIMENTS

In the following, we show the performance of MC-PNL in model learning and its application to bivariate causal discovery.

## 5.1 NONLINEAR FUNCTION FITTING

For better demonstration, we generated two synthetic datasets from the PNL model, $Y = f_2(f_1(X) + \epsilon)$, and each contains 1000 samples. The data generation mechanisms are as follows,

- Syn-1: $f_1(X) = X^{-1} + 10X, f_2(Z) = Z^3, X \sim U(0.1, 1.1), \epsilon \sim U(0, 5)$,
- Syn-2: $f_1(X) = \sin(7X), f_2(Z) = \exp(Z), X \sim U(0, 1), \epsilon \sim N(0, 0.3^2)$.

We apply Algorithm 1 to both datasets and show the learned nonlinear transformations as well as the corresponding residual plots in Figure 2. The underlying nonlinear functions are correctly learned under the true hypothesis but with certain distortions. We also show that, after fine-tuning with our proposed banded loss or HSIC-RBF loss, such distortion can be fixed up, see supplement H.

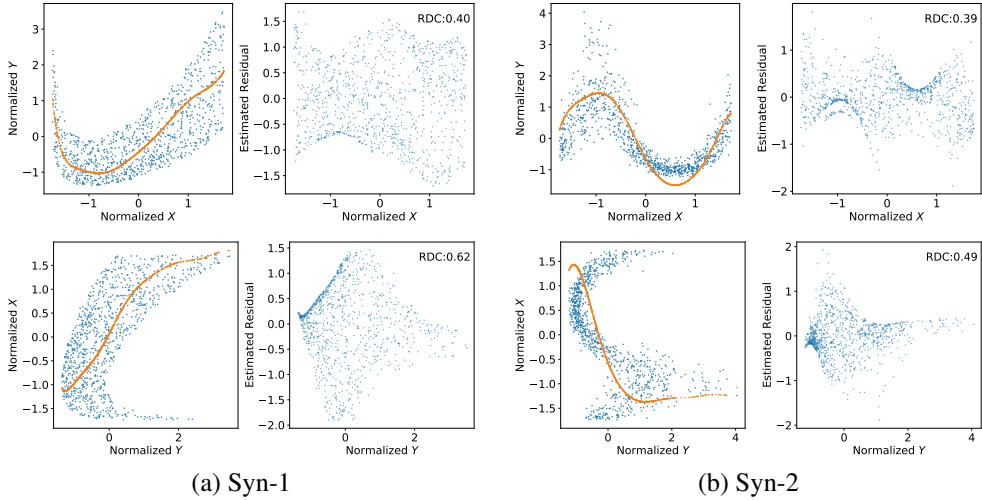

(a) Syn-1          (b) Syn-2

Figure 2: The sub-figures (a) and (b) show the nonlinear function fitting of the two datasets. In each sub-figure, the top row shows the learned $f_{(\rightarrow)}(x)$ (red line) and the residual plot under the correct hypothesis $X \rightarrow Y$, which has lower RDC value; the bottom row is under the opposite $X \leftarrow Y$.

**Convergence Results.** We demonstrate the convergence profile of our algorithm with Syn-2, see Figure 3. Results for Syn-1 can be found in the supplement J. The top row shows the snapshots of the learned representations, where we do not impose independence regularization ($\lambda = 0$). The algorithm, starting from different random initializations, convergences quickly to the local minimizers sharing the same objective value. The bottom row is with independence regularization $\lambda = 5$, where the solutions have a sign symmetry.

## 5.2 BIVARIATE CAUSAL DISCOVERY

We evaluated the causal discovery accuracy on both synthetic and real datasets.

**Synthetic Datasets:** The generated synthetic datasets all follow the PNL model. And we considered the following two settings: 1) PNL-A: $f_1$ are general nonlinear functions generated by polynomials with random coefficients; and $f_2$ are monotonic nonlinear functions generated by unconstrained monotonic neural networks (UMNN) (Wehenkel & Louppe, 2019); 2) PNL-B: Both $f_1$ and $f_2$ are monotonic generated by UMNN. The variances of $f_1, f_2$ are rescaled to 1. The input variable $X$ is sampled either from Gaussian mixture (mixG) or uniform (unif) distribution, and the injected noise $\epsilon$ is generated from normal distributions $N(0, \text{ns}^2)$, where $\text{ns} \in \{0.2, 0.4, 0.6, 0.8, 1\}$. Each configuration contains 100 data pairs, and each data pair has 1000 samples.

**Gene Data:** Discovering gene-gene causal relationships is one important application. We used the data in DREAM4 competition (D4-S1,D4-S2A,D4-S2B,D4-S2C) (Marbach et al., 2009; 2010) and the scRNA-seq data (GSE57872) (Han et al., 2017), see supplement K.

**Baselines & Evaluation:** Thanks to the implementation by Kalainathan et al. (2020), we can easily compare our proposed method with various existing algorithms. In this paper, we compared our

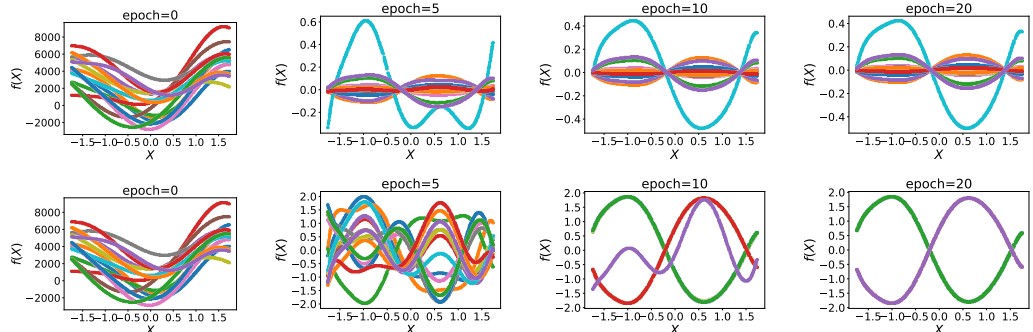

Figure 3: The Algorithm 1 converges on Syn-2. We plot the snapshots of the feature transformations $f$ at training epochs [0, 5, 10, 20], under 15 random initializations (indicated by colors). **Upper:** $\lambda = 0$, most initializations converge to local minimizers (symmetry: $(\boldsymbol{\alpha}, \boldsymbol{\beta}) \mapsto (a\boldsymbol{\alpha}, a^{-1}\boldsymbol{\beta})$). **Lower:** $\lambda = 5$, most initializations converge to two local minimizers (symmetry: $(\boldsymbol{\alpha}, \boldsymbol{\beta}) \mapsto -(\boldsymbol{\alpha}, \boldsymbol{\beta})$).

proposed algorithm on both synthetic datasets and real datasets with several baseline algorithms, including ANM (Hoyer et al., 2008), CDS (Fonollosa, 2019), IGCI (Janzing et al., 2012), RECI (Blöbaum et al., 2018), CDCI (Duong & Nguyen, 2022), OT-PNL (Tu et al., 2022), AbPNL (Uemura & Shimizu, 2020). Our implementation of MC-PNL follows Algorithm 1 (without fine-tuning), and we empirically set $\lambda = 5$ (the choice of $\lambda$ is briefly discussed in supplement L). We also conducted causal discovery on the PNL learned by the ACE algorithm. The ROC-AUC score is used for the evaluation.

Table 2: Comparison of bivaraite causal discovery ROC-AUC on synthetic and real datasets

| Dataset | ANM[1] | CDS | IGCI | RECI | CDCI | OT-PNL | AbPNL[1] | ACE[1] | MC-PNL[1] |
|---|---|---|---|---|---|---|---|---|---|
| PNL-A-mixG | 0.256 | 0.207 | 0.932 | 0.537 | 0.410 | 0.431 | 0.645 | 0.580 | 0.708 |
| PNL-B-mixG | 0.150 | 0.160 | 0.908 | 0.462 | 0.304 | 0.309 | 0.672 | 0.536 | 0.771 |
| PNL-A-unif | 0.203 | 0.390 | 0.681 | 0.879 | 0.544 | 0.711 | 0.517 | 0.514 | 0.617 |
| PNL-B-unif | 0.094 | 0.311 | 0.866 | 0.929 | 0.535 | 0.536 | 0.599 | 0.418 | 0.608 |
| D4-S1 | 0.604 | 0.582 | 0.380 | 0.550 | 0.651 | 0.474 | 0.408 | 0.592 | 0.646 |
| D4-S2A | 0.616 | 0.580 | 0.447 | 0.592 | 0.673 | 0.472 | 0.519 | 0.558 | 0.626 |
| D4-S2B | 0.521 | 0.529 | 0.450 | 0.491 | 0.614 | 0.517 | 0.501 | 0.495 | 0.519 |
| D4-S2C | 0.556 | 0.564 | 0.441 | 0.521 | 0.590 | 0.490 | 0.445 | 0.538 | 0.576 |
| GSE57872 | 0.493 | 0.457 | 0.599 | 0.474 | 0.478 | - | - | 0.538 | 0.499 |
| Average time[2] (s) | 20.11 | 7.67 | 0.50 | 0.27 | 0.26 | $\sim 7220$ | $\sim 9300$ | 21.68 | 30.31 |

[1] Independence test-based methods.
[2] Average running time evaluated on synthetic data containing 100 pairs, and each pair has 1000 samples.

We report the comparison of ROC-AUCs in Table 2. The results are averaged over five different noise scales for the synthetic datasets. Our proposed MC-PNL consistently outperforms other independence test-based methods on the synthetic PNL data. Especially compared with AbPNL, our MC-PNL is not sensitive to the initializations and is much more efficient (w.r.t. training time); compared with ACE (without independence regularizer), MC-PNL has better causal discovery accuracy. And for real datasets, our methods is quite competitive.

## 6 CONCLUSIONS

In this paper, we focus on the PNL model learning and propose a maximal correlation-based method, which can recover the nonlinear transformations accurately and swiftly in an iterative manner. The key is to incorporate with maximal correlation to avoid learning arbitrary independent noise, and the proposed MC-PNL is more reliable than previous methods that are solely based on the independence loss. Besides the PNL model learning, we conduct experiments on the downstream causal discovery task where MC-PNL is superior to the SOTA independence test-based methods.

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

# Supplementary Document
# Maximal Correlation-Based Post-Nonlinear Learning for Bivariate Causal Discovery

## A    DISCUSSION ON MI MINIMIZATION

It was shown that minimizing the MI in (5) is equivalent to maximizing $\mathbb{E} \log p\left(r_{(\rightarrow)}\right) +$ $\mathbb{E} \log \left|\frac{d}{dy} g_{(\rightarrow)}(y)\right|$ (Zhang & Hyvärinen, 2009), where $p$ is the assumed noise density. We find this objective interpretable, since the first term, $\mathbb{E} \log p\left(r_{(\rightarrow)}\right)$, can be understood as the data fitting term, and the second term, $\mathbb{E} \log \left|\frac{d}{dy} g_{(\rightarrow)}(y)\right|$, can be understood from an information-geometric perspective (Daniušis et al., 2010). However, such equivalent form requires a known noise distribution to calculate the log-likelihood. Some works (Ma et al., 2020; Uemura & Shimizu, 2020) have been proposed to avoid this difficulty by using HSIC instead of MI.

## B    EXPERIMENTS ON MINIMIZING HSIC

In this section, we show the PNL model learning result by minimizing (12). We generated two synthetic datasets from PNL model, $Y = f_2\left(f_1(X) + \epsilon\right)$, and each contains 1000 data samples. The data generation mechanisms are as follows (see Figure 4),

- Syn-1: $f_1(X) = X^{-1} + 10X, f_2(Z) = Z^3, X \sim U(0.1, 1.1), \epsilon \sim U(0, 5)$,
- Syn-2: $f_1(X) = \sin(7X), f_2(Z) = \exp(Z), X \sim U(0, 1), \epsilon \sim N(0, 0.3^2)$.

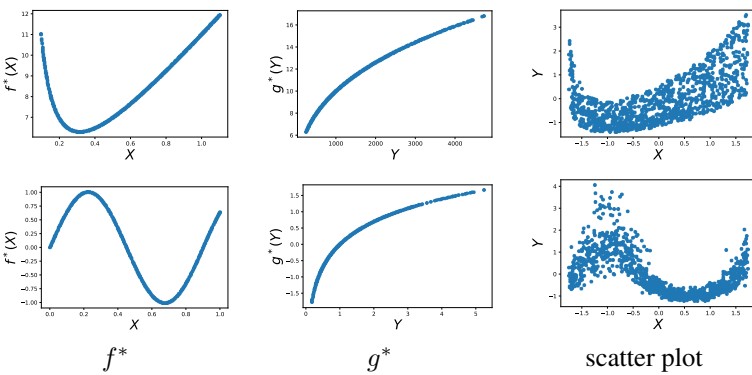

$f^*$ $\qquad\qquad\qquad$ $g^*$ $\qquad\qquad\qquad$ scatter plot

Figure 4: The ground truth transformations of $f^*$ and $g^*$ of Syn-1 (top) and Syn-2 (bottom).

We build different MLPs with the following configurations.

- Narrow deep MLP: the input and output are both one-dimensional; there are 9 hidden layers, each with 5 neurons. The activation function is `Leaky-ReLU`.
- Wide over-parameterized MLP: the input and output are both one-dimensional; there is only one single hidden layer with 9000 neurons. The activation function is `Leaky-ReLU`.

We use the default initialization method in PyTorch (Paszke et al., 2019), and make sure the exact same initial weights for narrow/wide MLPs are used (i.e., the initializations for different datasets are the same).

**Optimization Setup:** We set the batch size to be 32. We use `Adam` (Kingma & Ba, 2015) for the optimization (the learning rates are $10^{-3}$ and $10^{-6}$ for narrow deep and wide over-parameterized MLPs, respectively, while all other parameters are set by default).

We report the learning results in Figure 5. The learned transformations (see row 3 and row 4 in Figure 5) deviates far away from the underlying functions, and are quite similar across datasets. The

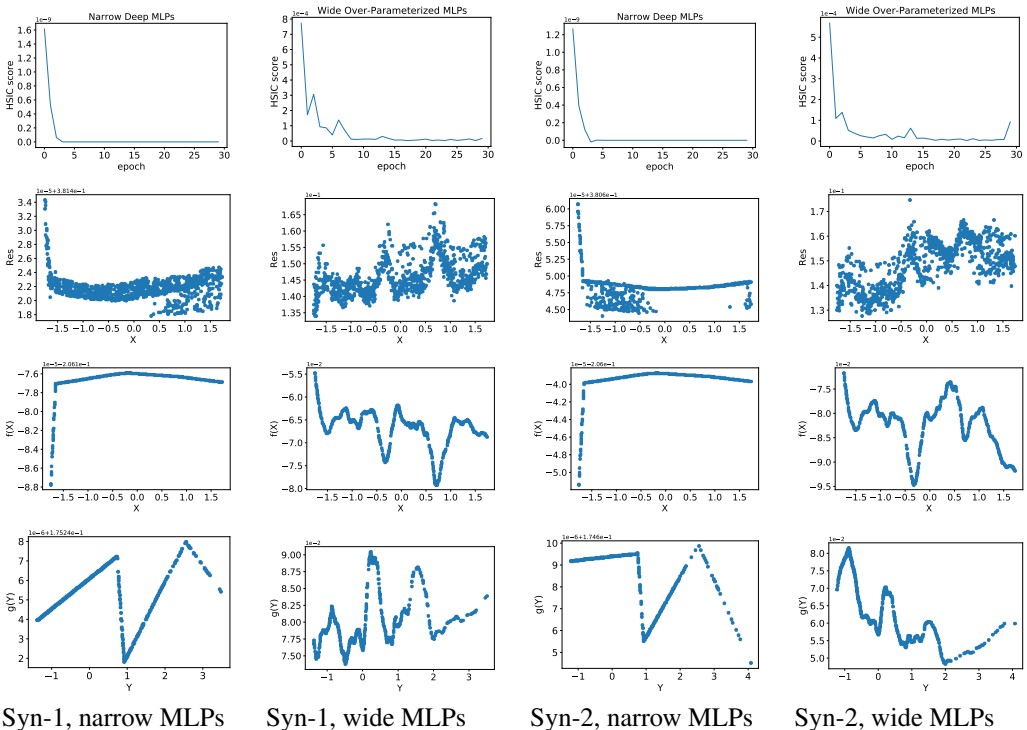

| Syn-1, narrow MLPs | Syn-1, wide MLPs | Syn-2, narrow MLPs | Syn-2, wide MLPs |

Figure 5: Visualization of the learned nonlinearities (trained solely with HSIC, under different datasets and MLP configurations). From top to bottom, the **convergence results**, **residual plot**, **learned** $f$, **learned** $g$, are plotted. Each column shows one specific configuration. None of them learns meaningful nonlinearities, and the learned transformations are quite similar across datasets.

possible reason is that, the solutions were started from the same initialization and trapped at the local minima near the initializations.

To verify whether such HSIC-based PNL learning algorithm is stable for causal discovery, we further evaluate the AbPNL on the following dataset. We build 100 data pairs with different random seeds, following the same mechanism, Syn-1, and each contains 1000 data samples. And we applied the AbPNL (Uemura & Shimizu, 2020) with different initializations on each of those data pairs. The results in Table 3 show that the causal discovery stableness for AbPNL is not satisfactory.

Table 3: Comparison of bivaraite causal discovery AUC on 100 realizations of Syn-1

| Dataset | ANM | CDS | IGCI | RECI | CDCI | AbPNL | ACE | MC-PNL |
|---------|-----|-----|------|------|------|-------|-----|--------|
| Syn-1 | 0.495 | 1 | 0.528 | 1 | 1 | 0.281 | 1 | 1 |

## C SYNTHETIC DATASETS FOR INDEPENDENCE TEST

In this section, we describe the synthetic data generation from PNL model for the independent test. The data were generated from the following model, $Y = f_2 \left( f_1(X) + \epsilon \right), X \sim \text{GMM}, \epsilon \sim N(0, \sigma_\epsilon^2)$, where $f_1, f_2$ are randomly initialized monotonic neural networks (Wehenkel & Louppe, 2019) with 3 layers and 100 integration steps, and each layer contains 100 units. The cause term $X$ is sampled from a Gaussian mixture model as described in Lopez-Paz et al. (2017). The datasets were configured with various noise levels and sample sizes. There are three different injected noise levels, $\sigma_\epsilon \in \{0.1, 1, 10\}$, and three different sample sizes, $N \in \{1000, 2000, 5000\}$. And under each configuration, we generated 100 data pairs for evaluating the independence test accuracy.

# D  A UNIVERSAL VIEW OF DEPENDENCE MEASURES

Actually the discussed dependence measures in Section 3.2 are all closely related to the *mean squared contingency* introduced by (Rényi, 1959) and rediscovered due to its squared version called *squared-loss mutual information* (SMI) (Suzuki et al., 2009),

$$\text{SMI} := \iint p(x)p(y) \left( \frac{p(x,y)}{p(x)p(y)} - 1 \right)^2 \mathrm{d}x\mathrm{d}y = \iint \frac{p(x,y)}{p(x)p(y)} p(x,y)\mathrm{d}x\mathrm{d}y - 1. \quad (16)$$

When the density ratio $\text{DR}(x,y) := \frac{p(x,y)}{p(x)p(y)}$ is constant 1 (namely $X$ and $Y$ are independent), the SMI should be zero. To estimate the SMI, one can first approximate $\text{DR}(x,y)$ by a surrogate function $\text{DR}_{\boldsymbol{\theta}}(x,y)$ parameterized by $\boldsymbol{\theta}$. The optimal parameter $\hat{\boldsymbol{\theta}}$ can be obtained via minimizing the following squared-error loss $J^{\text{DR}}$,

$$\begin{aligned} J^{\text{DR}}(\boldsymbol{\theta}) &:= \iint \left( \text{DR}_{\boldsymbol{\theta}}(x,y) - \text{DR}(x,y) \right)^2 p(x)p(y)\mathrm{d}x\mathrm{d}y \\ &= \iint \text{DR}_{\boldsymbol{\theta}}(x,y)^2 p(x)p(y)\mathrm{d}x\mathrm{d}y - 2 \iint \text{DR}_{\boldsymbol{\theta}}(x,y)p(x,y)\mathrm{d}x\mathrm{d}y + \text{Const.} \end{aligned} \quad (17)$$

Then the empirical SMI can be calculated as, $\widehat{\text{SMI}} = \frac{1}{n} \sum_{j=1}^{n} \text{DR}_{\hat{\boldsymbol{\theta}}}(x_j, y_j) - 1$.

We show that, with different parameterizations of the density ratio, the resulting SMI will be equivalent to different dependence measures, see Table 4.

Table 4: Connections between DR parameterization and dependence measure

| Density ratio surrogate function $\text{DR}_{\boldsymbol{\theta}}(x,y)$ | Corresponding dependence measure |
|---|---|
| $\text{DR}_{\boldsymbol{\theta}}(x,y) = 1 + \sum_{i=1}^{n} \theta_i K(x,x_i) L(y,y_i)$ | variant of LSMI (Sugiyama & Yamada, 2012) |
| $\text{DR}_{\boldsymbol{\theta}}(x,y) = 1 + \sum_{i=1}^{n} \frac{1}{n} K(x,x_i) L(y,y_i)$ | HSIC (Gretton et al., 2005) |
| $\text{DR}_{\boldsymbol{\theta}}(x,y) = 1 + \sum_{i=1}^{m} f_i(x)g_i(y)$ | $m$-mode HGR correlation (Wang et al., 2019) |
| $\text{DR}_{\boldsymbol{\theta}}(x,y) = 1 + f(x)g(y)$ [1] | HGR correlation (Rényi, 1959) |

[1] When $f,g$ are the linear combinations of random features, $f(x) = \boldsymbol{\alpha}^T \boldsymbol{\phi}(x), g(y) = \boldsymbol{\beta}^T \boldsymbol{\psi}(y)$, the corresponding dependence measure will be RDC (López-Paz et al., 2013),

Sugiyama & Yamada (2012) proposed to approximate the density ratio by $\text{DR}_{\hat{\boldsymbol{\theta}}}(x,y) = \sum_{i=1}^{n} \hat{\theta}_i K(x,x_i) L(y,y_i)$, where $\hat{\boldsymbol{\theta}}$ has a closed-form solution via minimizing (17). After then, they approximated the SMI using the empirical average of Equation (16), $\frac{1}{n} \sum_{j=1}^{n} \text{DR}_{\hat{\boldsymbol{\theta}}}(x_j, y_j) - 1 = \frac{1}{n} \sum_{j=1}^{n} \sum_{i=1}^{n} \hat{\theta}_i K(x,x_i) L(y,y_i) - 1$. It is shown that, the first term is actually the empirical HSIC, when $\{\hat{\theta}_i\}_{i=1}^{n} = \frac{1}{n}$. We argue that there is a flaw above, as when $X$ and $Y$ are independent, both the SMI and HSIC score should be zero. A simple modification is to model the density ratio by $\text{DR}_{\boldsymbol{\theta}}(x,y) = 1 + \sum_{i=1}^{n} \theta_i K(x,x_i) L(y,y_i)$. The constant 1 here is to exclude all the independence terms, and the rest ones should model the dependency only. This modification will not hurt the quadratic form of $J^{\text{DR}}(\boldsymbol{\theta})$, and maintains good interpretation. The SMI reduced to HSIC score, when $\{\theta_i\}_{i=1}^{n} = \frac{1}{n}$,

We extend this idea to approximate the density ratio by $\text{DR}_{\boldsymbol{\theta}}(x,y) = 1 + f(x)g(y)$, where $f,g$ are zero mean and unit variance functions parameterized by $\boldsymbol{\theta}$, the resulting SMI will be equal to the HGR maximal correlation. Similarly, the constant 1 will capture the independence part, and $f(x)g(y)$ will capture the dependencies.

**Proposition 1.** *The density ratio estimation problem (17) is equivalent to the maximal HGR correlation problem (7), when the density ratio is modeled in the form of $\text{DR}_{\boldsymbol{\theta}}(x,y) = 1 + f(x)g(y)$, and $f,g$ are restricted to zero mean and unit variance functions.*

*Proof.* We substitute $\mathrm{DR}_{\hat{\boldsymbol{\theta}}}(x, y)$ into Equation (17),

$$J^{\mathrm{DR}}(f, g) = \iint (1 + f(x)g(y))^2 p(x)p(y)\mathrm{d}x\mathrm{d}y - 2 \iint (1 + f(x)g(y))p(x, y)\mathrm{d}x\mathrm{d}y + \mathrm{Const.}$$

$$= 1 + 2\mathbb{E}(f(X))\mathbb{E}(g(Y)) + \mathrm{var}(f(X))\mathrm{var}(g(Y)) - 2 - 2\mathbb{E}(f(X)g(Y)) + \mathrm{Const.}$$

Then it is not hard to see, $\min_{f,g} J^{\mathrm{DR}}(f, g)$, subject to $\mathbb{E}(f) = \mathbb{E}(g) = 0, \mathrm{var}(f) = \mathrm{var}(g) = 1$, is equivalent to the maximal HGR correlation problem (7) . $\qquad\square$

**Proposition 2.** *The density ratio estimation problem (17) is equivalent to the Soft-HGR problem (9), when the density ratio is modeled in the form of* $\mathrm{DR}_{\boldsymbol{\theta}}(x, y) = 1 + f(x)g(y)$, *and* $f, g$ *are restricted to zero mean functions.*

We further note that the above density ratio estimation can be regard as a truncated singular value decomposition $\mathrm{DR}_{\hat{\boldsymbol{\theta}}}(x, y) = 1 + \sum_{i=1}^{m} f_i(x)g_i(y)$, where $m = 1$. When letting $m > 1$ and imposing zero mean and unit variance constraints on all $f_i$ and $g_i$, the corresponding $J^{\mathrm{DR}}$ minimization problem is equivalent to solving the $m$-mode HGR maximal correlation (Wang et al., 2019; Lee, 2021).

**Definition 3** ($m$-mode HGR maximal correlation). *Given* $1 \leq m \leq \min\{|\mathcal{X}|, |\mathcal{Y}|\}$, *the* $m$-mode *maximal correlation problem for random variables* $X \in \mathcal{X}, Y \in \mathcal{Y}$ *is,*

$$(\mathbf{f}^*, \mathbf{g}^*) \triangleq \underset{\substack{\mathbf{f}:\mathcal{X}\to\mathbb{R}^m, \mathbf{g}:\mathcal{Y}\to\mathbb{R}^m \\ \mathbb{E}[\mathbf{f}(X)]=\mathbb{E}[\mathbf{g}(Y)]=\mathbf{0}, \\ \mathbb{E}[\mathbf{f}(X)\mathbf{f}^{\mathrm{T}}(X)]=\mathbb{E}[\mathbf{g}(Y)\mathbf{g}^{\mathrm{T}}(Y)]=\mathbf{I}}}{\arg\max} \mathbb{E}\left[\mathbf{f}^{\mathrm{T}}(X)\mathbf{g}(Y)\right], \tag{18}$$

*where* $\mathbf{f} = [f_1, f_2, \ldots, f_m]^{\mathrm{T}}, \mathbf{g} = [g_1, g_2, \ldots, g_m]^{\mathrm{T}}$ *are referred as the maximal correlation functions.*

## E    CONNECTIONS AMONG MI, ML, AND MC

In this section, we build connections among minimizing MI, maximum likelihood, and maximal correlation. The equivalence between minimizing MI and maximizing likelihood was built in Zhang & Hyvärinen (2009). The following proposition shows the connection to maximal correlation.

**Proposition 3.** *Suppose the dataset* $\{(x_i, y_i)\}_{i=1}^n$ *is generated from a PNL model* $Y = g^{-1}(f(X) + \epsilon)$, *where* $f, g$ *are both* ***invertible functions***, *and the noise* $\epsilon$ *follows a Gaussian density* $p(\epsilon; \theta)$ *with zero mean and variance* $\theta$, *then maximizing the log-likelihood* $\log p(\{(x_i, y_i)\}_{i=1}^n)$ *is equivalent to solving the regression problem (8).*

*Proof.* Under proper assumptions in proposition 3, the log-likelihood can be written as follows,

$$\begin{aligned} L_n(f, g) &= \sum_{i=1}^n \log p(x_i, y_i; f, g, \theta) \\ &= \sum_{i=1}^n \log p(x_i) + \sum_{i=1}^n \log p(y_i|x_i; f, g, \theta) \\ &= \sum_{i=1}^n \log p(x_i) + \sum_{i=1}^n \log p(g(y_i)|f(x_i); \theta) \quad (f, g \text{ are invertible}) \\ &= \sum_{i=1}^n \log p(x_i) + \sum_{i=1}^n \log p(g(y_i) - f(x_i); \theta) \quad (\text{from PNL model}) \\ &= \sum_{i=1}^n \log p(x_i) + \sum_{i=1}^n \frac{-(g(y_i) - f(x_i))^2}{2\theta} + n \log \frac{1}{\sqrt{2\pi\theta}} \quad (\text{Gaussianity}) \quad (19) \end{aligned}$$

It is not hard to see, with fixed $\theta$, maximizing the log-likelihood $L_n(f, g)$ is equivalent to minimizing $\|f(\mathbf{x}) - g(\mathbf{y})\|^2$ with invertible $f$ and $g$. Without loss of generality, one can make $f, g$ zero mean. To avoid trivial solutions, one can further restrict $g$ to have unit-variance. Then the equivalence to the regression problem (8) is build. $\qquad\square$

**Corollary 1.** *When $n \to \infty$, the ground truth transformations $f^*, g^*$, minimize the $\mathrm{MI}(\mathbf{x}, \hat{\mathbf{r}})$ to zero, achieve optimum of (8), and maximize the log-likelihood $L_n(f^*, g^*)$.*

*Proof.* The proof is directly follows Theorem 3 in (Zhang et al., 2015). $\square$

The reformulating to (8) or (9)[1] allows efficient BCD-like optimization algorithms to be exploited.

## F RANDOM FEATURE GENERATION

We generate the random features as described in López-Paz et al. (2013). The generation process has the following two steps: *copula transformation* (optional) and *random nonlinear projection*.

**Step 1.** *Copula transformation.* We first estimate the empirical cumulative distribution of both $X$ and $Y$ by,

$$P_n^X(x) := \frac{1}{n} \sum_{i=1}^n \mathbb{I}\left(x_i \leq x\right), P_n^Y(y) := \frac{1}{n} \sum_{i=1}^n \mathbb{I}\left(y_i \leq y\right).$$

Then we can apply the empirical copula transformation to data samples $\{(x_i, y_i)\}_{i=1}^n$, $u_i^X = P_n^X(x_i)$ and $u_i^Y = P_n^Y(y_i)$, where the marginals $U^X$ and $U^Y$ follow uniform distribution $U(0,1)$.

**Step 2.** *Random nonlinear projection.* We design a $k$-dimensional random feature vector $\phi(x) = [\sin(w_1 x + b_1), \cdots, \sin(w_k x + b_k)]^T$, where $w_i, b_i \sim N(0, s^2)$. The random feature matrix $\Phi \in \mathbb{R}^{k \times n}$ is stacked as,

$$\Phi(\mathbf{x}; k, s) := \left( \begin{array}{ccc} \sin\left(w_1 x_1 + b_1\right) & \cdots & \sin\left(w_1 x_n + b_1\right) \\ \vdots & \vdots & \vdots \\ \sin\left(w_k x_1 + b_k\right) & \cdots & \sin\left(w_k x_n + b_k\right) \end{array} \right).$$

One can replace the $x_i$ here by $u_i^X$ from the first step to form the random feature matrix. Similar procedures can be applied to $\mathbf{y}$ as well to generate $\Psi$. The number of random Fourier features $k$ is user-defined, which is typically chosen from a few tens to a few thousands (Rahimi & Recht, 2008; Theodoridis, 2015). In our experiments, we set $k = 30$ and $s = 2$.

## G ON THE OPTIMIZATION OF PROBLEM (14)

### G.1 SUBPROBLEM: EQUALITY CONSTRAINED QUADRATIC PROGRAMMING

To simplify the notation, we rewrite the sub-problem into the following form,

$$\begin{aligned} \min_{x \in \mathbb{R}^n} \quad & f(x) := \tfrac{1}{2} x^T A x - b^T x, \\ s.t. \quad & v^T x = c. \end{aligned} \tag{20}$$

With the KKT conditions, one can find the unique optimal solution $x^*$ by solving the following linear system,

$$\underbrace{\left( \begin{array}{cc} A & v \\ v^T & 0 \end{array} \right)}_{=:\mathrm{KKT}} \left( \begin{array}{c} x^* \\ \lambda^* \end{array} \right) = \left( \begin{array}{c} b \\ c \end{array} \right), \tag{21}$$

when the KKT matrix is non-singular. In our setting, we can choose $\Phi$ and $\Psi$ properly to make $\Phi\Phi^T$ and $\Psi\Psi^T$ positive definite, or add a small positive definite perturbation matrix $\epsilon I$, such that the unique optimum would be obtained. Besides, the sub-problem is of smaller size and easy to solve.

---

[1] We note that the optimal solution of (8) is also one solution of (9).

## G.2 LANDSCAPE STUDY WITH HESSIAN

To simplify the notation, we rewrite

$$J(\boldsymbol{\alpha}, \boldsymbol{\beta}; A, B, C, D, E) = \boldsymbol{\alpha}^T A \boldsymbol{\alpha} \boldsymbol{\beta}^T B \boldsymbol{\beta} - \boldsymbol{\alpha}^T C \boldsymbol{\beta} + \boldsymbol{\alpha}^T D \boldsymbol{\alpha} + \boldsymbol{\beta}^T E \boldsymbol{\beta}, \tag{22}$$

where,

$$\begin{aligned}
A &= \tfrac{1}{2n^2} \Phi \Phi^T, \\
B &= \Psi \Psi^T, \\
C &= \tfrac{1}{n} \Phi \Psi^T + \tfrac{\lambda}{(n-1)^2} \Phi H K_{\mathbf{xx}} H \Psi^T, \\
D &= \tfrac{\lambda}{(n-1)^2} \Phi H K_{\mathbf{xx}} H \Phi^T, \\
E &= \tfrac{\lambda}{(n-1)^2} \Psi H K_{\mathbf{xx}} H \Psi^T.
\end{aligned} \tag{23}$$

And the corresponding Hessian is

$$\nabla^2 J(\boldsymbol{\alpha}, \boldsymbol{\beta}) = \begin{pmatrix} 2A\boldsymbol{\beta}^T B \boldsymbol{\beta} + 2D & A\boldsymbol{\alpha}\boldsymbol{\beta}^T B - C \\ B^T \boldsymbol{\beta}\boldsymbol{\alpha}^T A - C^T & 2B\boldsymbol{\alpha}^T A \boldsymbol{\alpha} + 2E \end{pmatrix}. \tag{24}$$

Now we are able to verify the property of the critical points via checking their Hessians numerically.

One obvious critical point is the all zero vector $\mathbf{0}$. From our experiments, the Hessian at $\mathbf{0}$ is mostly indefinite, as long as the convex regularization term $\lambda$ is not too huge, which means $\mathbf{0}$ is a saddle point. In practice, the algorithm rarely converges to $\mathbf{0}$.

## H  FINE-TUNE WITH BANDED LOSS / UNIVERSAL HSIC

In the PNL model, the injected noise are assumed to be independently and identically distributed. Thus, the residual plot should forms a "horizontal band". We design a **banded residual loss** to fine-tune the models as follows. The data samples are separated into $b$ bins $\{\mathbf{x}^{(i)}, \mathbf{y}^{(i)}\}_{i=1}^b$ according to the ordering of $X$, and we expect the residuals in those bins $\mathrm{Res}_i = f(\mathbf{x}^{(i)}) - g(\mathbf{y}^{(i)})$ to have the same distribution, see Figure 6. To this end, we adopt the empirical maximum mean discrepancy (MMD) (Gretton et al., 2012) as a measure of distribution discrepancy. The **banded residual loss** is defined as $\mathrm{band}^{(\mathrm{MMD})} := \sum_{i=1}^b \widehat{\mathrm{MMD}}(\mathrm{Res}_i, \mathrm{Res}_{all})$, where $\mathrm{Res}_{all} = f(\mathbf{x}) - g(\mathbf{y})$. Then we append this $\mu$-penalized banded loss to Problem (14) as,

$$\min_{\boldsymbol{\alpha}, \boldsymbol{\beta}} \quad J(\boldsymbol{\alpha}, \boldsymbol{\beta}) + \mu \cdot \mathrm{band}^{(\mathrm{MMD})}, \quad \text{s.t.} \quad \boldsymbol{\alpha}^T \Phi \mathbf{1} = \boldsymbol{\beta}^T \Psi \mathbf{1} = 0. \tag{25}$$

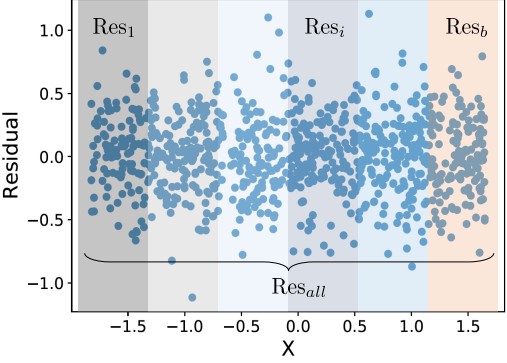

Figure 6: The construction of banded residual loss.

The above banded residual loss involves MMD, which is highly non-convex and brings difficulties to the optimization. We used the projected gradient descent with momentum to optimize the loss function. The residual plot shows a band shape, see top row in Figure 7.

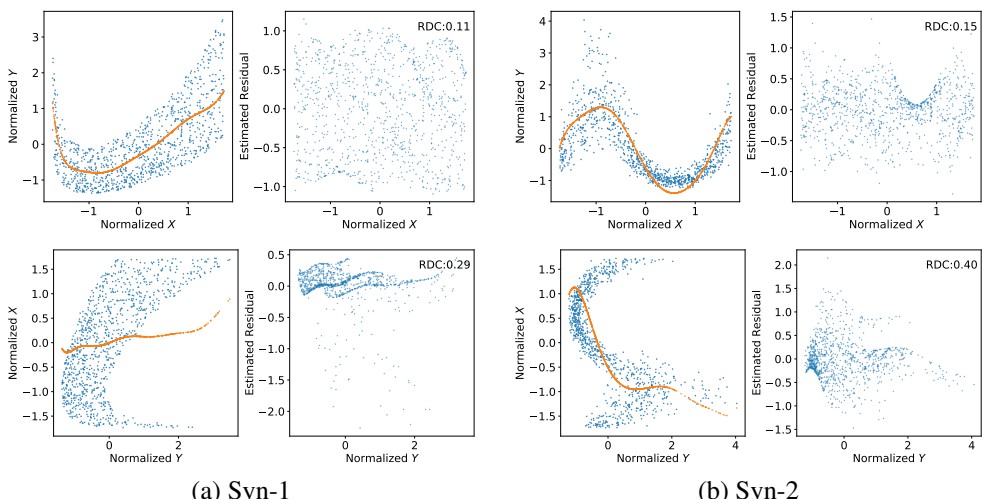

Figure 7: Fine-tuning with the banded residual loss.

We also show the results of fine-tuning by enlarging the penalty (to $\lambda = 10000$) HSIC term with universal Gaussian RBF kernel in Figure 8.

**Definition 4** (Universal Kernel (Gretton et al., 2005))*. A continuous kernel $k(\cdot, \cdot)$ on a compact metric space $(\mathcal{X}, d)$ is called universal if and only if the RKHS $\mathcal{F}$ induced by the kernel is dense in $C(\mathcal{X})$, the space of continuous functions on $\mathcal{X}$, with respect to the infinity norm $\|f - g\|_\infty$.*

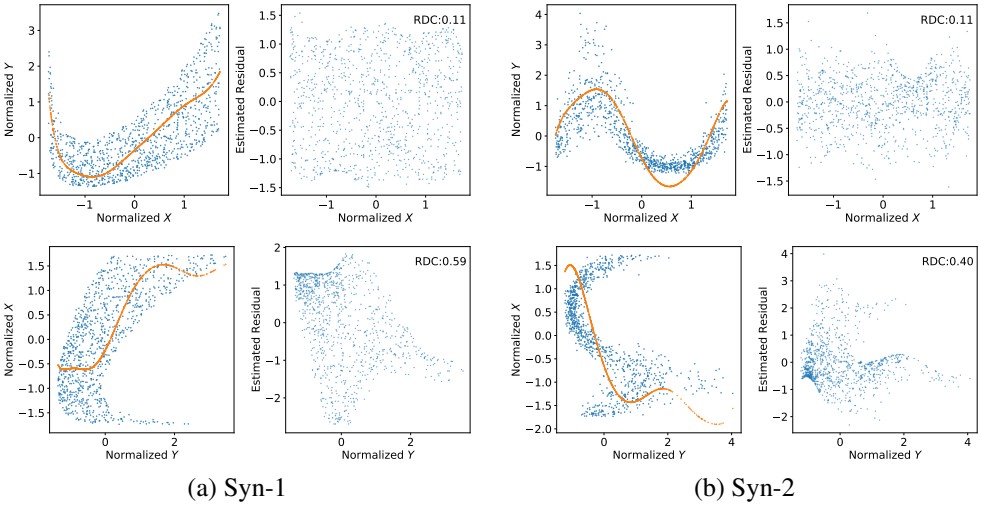

Figure 8: Fine-tuning with the HSIC-RBF loss.

# I  BOOTSTRAP FOR TRUSTWORTHY CAUSAL DISCOVERY

Bootstrap is a commonly used technique to estimate the confidence interval. In this section, we show a few examples of bootstrap with Tuebingen data (Mooij et al., 2016). We obtained 30 estimates of RDC from the data re-sampled with replacement, see Figure 9. The blue bars indicate the RDC distribution under the true causal direction; The orange bars indicate the RDC distribution under the false causal direction.

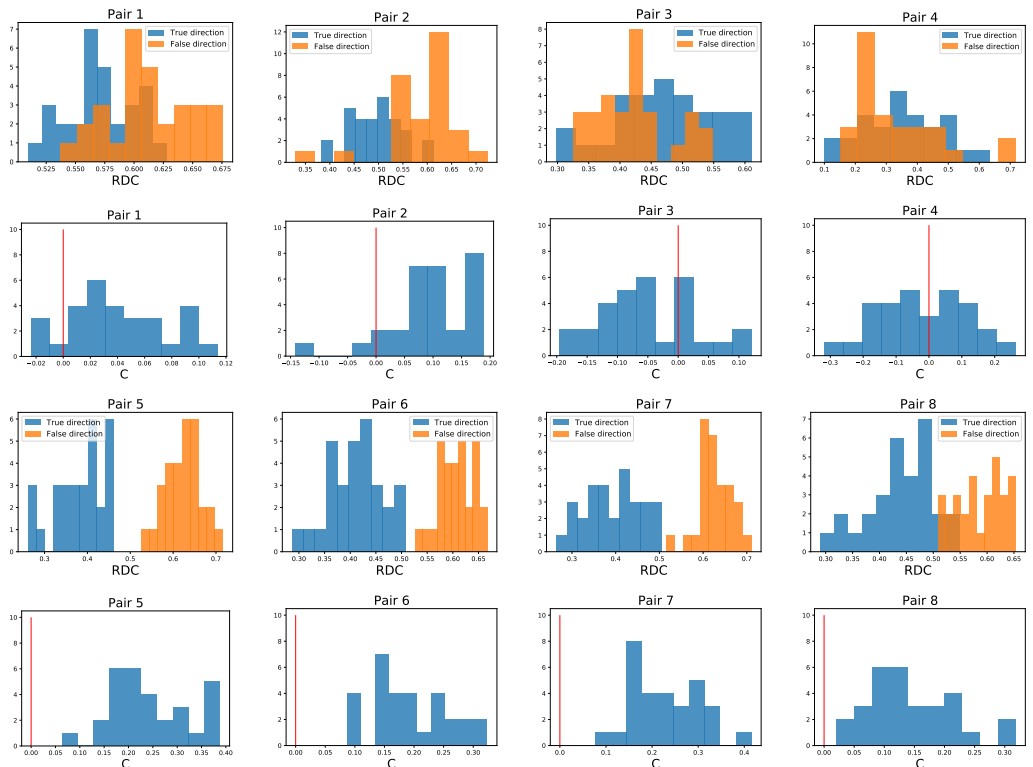

Figure 9: Bootstrap results of MC-PNL on eight Tuebingen datasets. We plot the histogram of the RDC estimates and the estimated causal scores of 30 replications.

## J  ADDITIONAL CONVERGENCE RESULTS

In this section, we show the convergence results on Syn-1 as well.

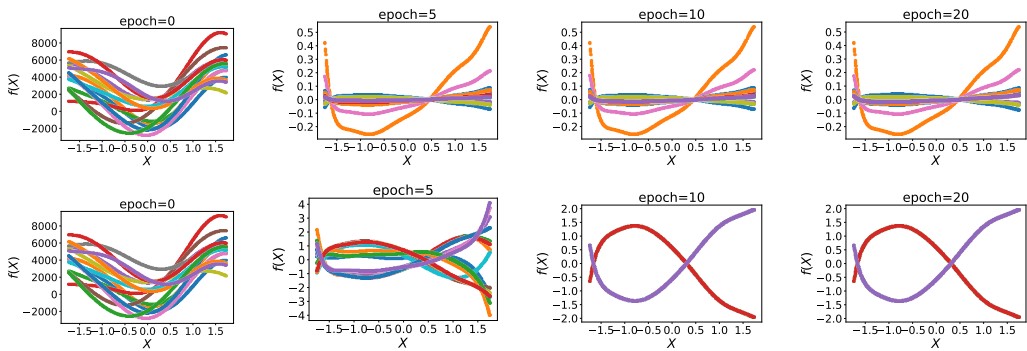

Figure 10: The Algorithm 1 converges on Syn-1. We plot the snapshots of the feature transformations $f$ at training epochs [0, 5, 10, 20], under 15 random initializations (indicated by colors). **Upper:** $\lambda = 0$, most initializations converge to local minimizers (symmetry: $(\boldsymbol{\alpha}, \boldsymbol{\beta}) \mapsto (a\boldsymbol{\alpha}, a^{-1}\boldsymbol{\beta})$). **Lower:** $\lambda = 5$, most initializations converge to two local minimizers (symmetry: $(\boldsymbol{\alpha}, \boldsymbol{\beta}) \mapsto -(\boldsymbol{\alpha}, \boldsymbol{\beta})$).

## K  DETAILED DATA DESCRIPTIONS

In this section, we describe the datasets in detail.

**Gene Datasets:**

For D4-S1, D4-S2A, D4-S2B, D4-S2C, we used the preprocessed data in Duong & Nguyen (2022) [2]. D4-S1 contains 36 variable pairs with 105 samples in each pair; D4-S2A, D4-S2B, D4-S2C contains 528, 747, and 579 variable pairs respectively, and each pair contains 210 samples.

The GSE57872 dataset is built on Patel et al. (2014), in which the data has continuous values. Following Choi et al. (2020), we first screen out 657 gene pairs that have corresponding labels in the TRRUST database (Han et al., 2017). The gene contains many repeated values. we examined each gene pair and deleted those repeated expression values.

## L    ON THE CHOICE OF $\lambda$

We tried seven different values for $\lambda$, and report the AUC scores on the PNL-A-unif dataset with different noise levels. We found that the MC-PNL is suitable to use in the small noise regime. We also found that for the data with small noise, smaller $\lambda$ is preferred; and for the data with large injected noise, larger $\lambda$ is preferred.

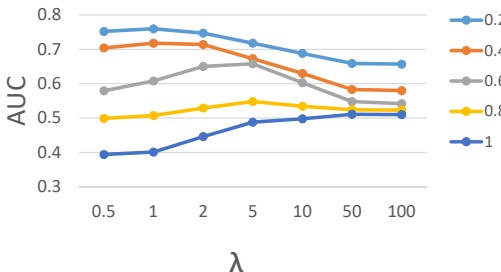

Figure 11: The detailed AUC scores vs. $\lambda$ under five noise levels on PNL-A-unif data.

---

[2]https://github.com/baosws/CDCI

