# OpenReview forum: "Maximal Correlation-Based Post-Nonlinear Learning for Bivariate Causal Discovery"
_ICLR.cc/2023/Conference — Submitted to ICLR 2023_

### Official Review · Reviewer_E1Jp · 2022-10-18

**Confidence:** 4
**Correctness:** 3
**Technical Novelty And Significance:** 3
**Empirical Novelty And Significance:** 3
**Recommendation:** 6

**Clarity, Quality, Novelty And Reproducibility:**

The paper is well-written.
There is some novelty in the contribution.

**Strength And Weaknesses:**

Strengths. The paper is in general well-written. It does a good overview of the state-of-the art methods. The proposed method is novel and worth being studied.

Weaknesses. It is stated that the proposed method outperforms the state-of-the-art independence test-based methods (bivariate causal inference), however, the method is much worse than some other state-of-the-art causal discovery methods. The results of the numerical experiments are not so convincing. I see that the idea is to show that the method outperforms the test-based methods, however, the IGCI method which is also based on the same asymmetry (independence of mechanism) hypothesis performs much better. Do you have some intuition? Are the independence tests used not efficient enough? And the future work should focus on a more efficient tests?

Figure 2 is unclear. What is the true causal direction? And how the plots should be interpreted? Taking, e.g., subplot a), above and below, we clearly see that the red line is well-fitted. What is the direction? And what can be deduced from the estimated residual values?

Algorithm 2 abstains from the decision only if C = 0. It is not too strict for real (noisy) applications? Were it be not better to introduce some \epsilon, so that if abs(C) < \epsilon the decision can not be made? As it is done, e.g. in F. Liu and L. Chan,“Causal inference on discrete data via estimating distance correlations,” Neural Computation, vol. 28, 2016.

Finally, in the problem formulation (equation 13), there are two parts: the HGR correlation and RDC (instead of HSIC which is, however, seems to be used in the paper). Could you study the efficiency of each of these parts?

Section 2.2: generating process and the causalities: I guess a number of assumptions are made behind the description of the idea, and these assumptions need to be mentioned.

In the numerical experiments (Section 5.2.) did you fix C = 0 (as in the Algorithm 2)?

While reading, I have an impression that the HSIC takes too much room in the paper, finally the HSIC is criticised and is proposed to be replaced.



**Summary Of The Paper:**

The paper proposes a new bivariate causal inference algorithm called MC-PNL (maximal correlation with independence regularisation). The authors propose to use the randomised dependence coefficient (RDC) instead of HSIC test often used by the causal inference community.


**Summary Of The Review:**

Some points need to be clarified.
I have some concerns about the numerical experiments, since the reported results outperform some existing methods but are much worse compared to some other state-of-the-art approaches.
A number of assumptions made in the theoretical part are not mentioned explicitly.

---

> ### Author Response · Authors · 2022-11-16
> **To Reviewer E1Jp**
>
> Thanks very much for your careful reading and comments.
>
> **Explanations on IGCI performance**
>
> We agree the discussion on IGCI would be beneficial to broader audiences. A brief implementation of IGCI can be found in [1], and the implementation adopted in our paper is entropy-based, i.e., $H(X)>H(Y) => X \rightarrow Y$. The IGCI does not require any independence test. One can see that the performance of IGCI is not consistently good across all the datasets, possibly due to the variance of sample size (in gene data, the sample size is pretty small).
>
> **Explanation of Figure 2**
>
> The data are generated according to the true causal direction, X->Y. Taking subplot (a), for example, the upper row is the fitting under the correct model hypothesis, so the red line is well-fitted, and the corresponding residual shows the independence with the input (of band shape, and the corresponding RDC is indicated as well); however, in the other direction (wrong direction), the fitting is bad, and the residual is not of the band shape.
>
> **C=0 is too strict?**
>
> Yes, we adopt this presentation of decision rule following the bivariate causal discovery framework proposed in [1]. As you suggested, one can make no decision when C is in a certain interval. Another typical remedy is to use bootstrap, which can give an empirical distribution of our RDC estimator and conduct hypothesis tests. We have added this comment after Algorithm 2.
>
> **Study two parts in (13), HGR correlation and RDC.**
>
> We actually have done such an ablation study. To see the effectiveness of involving maximal HGR correlation, we showed in supplement B that minimizing HSIC merely will get stuck around the random initialization, but combining both objectives can provide meaningful solutions. On the other hand, we also compared the results obtained using the maximal HGR correlation via the ACE algorithm (see ACE in Table 2) to show the effectiveness of involving the dependence penalty.
>
> **Assumptions of PNL generation in Section 2.2.**
>
> Two assumptions have been made for the PNL model: A1, independent noise $\epsilon$; A2, invertible post-nonlinear function f2.
>
> **In the numerical experiments (Section 5.2.) did you fix C = 0?**
>
> In our numerical experiments, we used the AUC to measure the performance. (Unlike ACC, which requires a fixed threshold value.)
>
> **“HSIC takes too much room in the paper, finally the HSIC is criticised and is proposed to be replaced.”**
>
> HSIC is an important component of our method. Directly using it for an independence test in causal discovery may be problematic, as it depends on the scale of random variables. However, using it in a loss function is fine. The HSIC loss is easy to evaluate and differentiable. We also use the HSIC loss in the PNL model learning.
>
>
>
>
> [1] Mooij, J. M., Peters, J., Janzing, D., Zscheischler, J., & Schölkopf, B. (2016). Distinguishing cause from effect using observational data: methods and benchmarks. The Journal of Machine Learning Research, 17(1), 1103-1204.

---

### Official Review · Reviewer_2uGF · 2022-10-24

**Confidence:** 3
**Correctness:** 3
**Technical Novelty And Significance:** 2
**Empirical Novelty And Significance:** 2
**Recommendation:** 6

**Clarity, Quality, Novelty And Reproducibility:**

The idea of combining the Soft-HCR and dependence measure (HSIC) is non-trivial. This paper is well-written.

**Strength And Weaknesses:**

Strength

The authors focus on the challenge of causal direction inferring. This is an important but challenging problem.

The paper builds on prior work in the field of casual discovery in PNL model. The authors put some effort into mitigating the problem of time-consuming and unreliable results with finite samples.

The basic idea is interesting and useful.

 This paper is well-written and well-organized.


Weakness

Some results are based on simulation analysis rather than theoretical analysis.


Some concerns or questions

1. The last but one paragraph on Page 2: a biased HSIC? It may be unbiased?

2. ﻿In Table 2, the ROC-AUCs of IGCI have better accuracy than the proposed method in the synthetic datasets. However, IGCI has worse accuracy in real-data sets. Can you explain this result?


**Summary Of The Paper:**

This paper studies bivariate causal direction learning from observational data. The authors focus on the general bivariate model, i.e., the post-nonlinear model, and propose a new method to learn the model. The contributions are as follows, the authors first analyze the drawbacks of the existing estimation methods, e.g., the PNL-MLP algorithm, and the AbPNL algorithm. Then they propose a new optimized function to estimate the PNL model by using the Soft-HCR. Finally, the authors apply their methods in both synthetic and Gene datasets to verify the efficiency.

**Summary Of The Review:**

The authors make some progress on the hard problem of estimating the causal direction of interest only using the observed variables.  The proposed method is non-trivial and useful with finite samples. However, some results are based on simulation analysis rather than theoretical analysis.

---

> ### Author Response · Authors · 2022-11-16
> **To Reviewer 2uGF**
>
> Thanks for your careful reading and comments.
>
> **Theoretical improvement:**
> As suggested by Reviewer 7wPL, we have included the following theoretical justifications in the revised version.
>
> - We show the connections of minimizing the mutual information, maximum likelihood, and maximal correlation under the assumptions of Gaussian noise and invertible transformations, f1 and f2.
> -  Under the above assumptions, the ground truth functions can achieve the optimum of (13).
>
>
> To address your concerns:
>
> - 1. $\frac{1}{n^2} tr(KHLH)$ is a **biased** estimator, see (4) in [1].
> - 2. **Explanation of IGCI performance.**
> The performance drop of IGCI may be due to the smaller **sample size**. The adopted IGCI implementation is based on an entropy estimator, which requires a relatively large dataset. In the simulation dataset, the number of samples is 1000; while in gene data, the number of samples is just a few hundreds, which can be too small for constructing a good entropy estimator.
>
> [1] Gretton, A., Fukumizu, K., Teo, C., Song, L., Schölkopf, B., & Smola, A. (2007). A kernel statistical test of independence. Advances in neural information processing systems, 20.

---

### Official Review · Reviewer_9qg4 · 2022-10-25

**Confidence:** 5
**Correctness:** 3
**Technical Novelty And Significance:** 2
**Empirical Novelty And Significance:** 2
**Recommendation:** 3

**Clarity, Quality, Novelty And Reproducibility:**

Presentation is clear in my opinion.

Novelty is not very high, as commented above.

Proposed methods seem reproducable.

**Strength And Weaknesses:**

The practical take-aways such as the strength of RCD with finite samples could be useful.

Authors have not tested their algorithm on Tuebingen dataset, which is the baseline for bivariate causal discovery.

**Summary Of The Paper:**

The authors suggest using the randomized dependence coefficient (RDC) instead of the Hilbert-Schmidt independence criterion (HSIC) for the independent test for performing bivariate causal discovery using MLPs in the post-nonlinear setting.

**Summary Of The Review:**

Proposing using a different independence measure is not a sufficient contribution in my opinion. This is analogous to having a different algorithm around the PC algorithm by simply using different conditional independence testers. Please see below for some more minor comments.

On Justification of the Method:
"However, the state-of-the-art (SOTA) PNL-based algorithms involve highly non-convex objectives for neural network training, which are
time-consuming and unable to produce meaningful solutions with finite samples."
Causal discovery literature does not generally rely on neural network training. So I find this justification a bit inadequate. How about fitting a function class then checking residual independence as is typically done with ANM models?

"moreover, the discovered DAG may not necessarily be causal."
This is not a fair criticism either. The experts are aware that GES would return a graph in the MEC.

"In this paper, we will focus on a more fundamental problem"
the relative fundamental-ness of bivariate vs. full graph discovery is subjective. I would suggest authors refrain from such subjective comparative statements.

On Experiments:

A typical benchmark used for bivariate causal discovery is the Tuebingen dataset. Did you test your algorithm on this real data? It would be nice to address why this was not added.

---

> ### Author Response · Authors · 2022-11-16
> **To Reviewer 9qg4**
>
> Thanks for your time to review our paper.
>
> It seems you have missed the major contributions of our paper. Suggesting RDC as a dependence measure is a very minor contribution of this work.
>
> **The meat** is the new proposed **maximal correlation-based** PNL learning method.
> When dealing with finite sample datasets, other PNL-based methods(including PNL-MLP and AbPNL) may be stuck in a bad local minimum near their initialization, producing meaningless results. However, our method can learn **meaningful and interpretable feature transformations**. The proposed **objective is benign and can be optimized efficiently** with the BCD algorithm.
>
> Regarding some minor points:
> - **"However, the state-of-the-art (SOTA) PNL-based algorithms involve highly non-convex objectives for neural network training, which are time-consuming and unable to produce meaningful solutions with finite samples."**
>
>      This claim specifies the subject to the PNL-based algorithms, and most of them are based on neural networks, typically with a non-convex loss (e.g., HSIC). We rephrased this sentence as follows.
>
>
>     "However, the state-of-the-art (SOTA) PNL-based algorithms involve highly non-convex objectives due to the use of neural networks and non-convex losses, thus optimizing such objectives is often time-consuming and unable to produce meaningful solutions with finite samples."
>
> - **"moreover, the discovered DAG may not necessarily be causal."** We insist on this claim: GES would return a graph in the MEC, but not every graph in the MEC is a causal graph.
> - **"In this paper, we will focus on a more fundamental problem"** => the word “more” is deleted.
>
> **On experiments:**
>
> - **Tuebingen data:** we have noticed this important benchmark dataset early on. In many literatures [1,2], people preferred to use a subset of this dataset for their evaluation. However, we could not find documents of these subsets choices to reproduce their results. So for fairness, we did not report the corresponding results in our paper.  Here, we provide a comparison on Tuebingen data version 1.0. We adopted the implementation of ANM, CDS, IGCI, and RECI, in the causal discovery tool package. On this dataset, we can see MC-PNL achieves the same result as AbPNL, with much lower computational complexity.
>
> |     | ANM   | CDS   | IGCI  | RECI  | CDCI  | AbPNL | ACE   | MC-PNL |
> |-----|-------|-------|-------|-------|-------|-------|-------|--------|
> | AUC | 0.552 | 0.667 | 0.657 | 0.705 | 0.722 | 0.567 | 0.537 | 0.567  |
>
> [1] Zhang, K., Wang, Z., Zhang, J., & Schölkopf, B. (2015). On estimation of functional causal models: general results and application to the post-nonlinear causal model. ACM Transactions on Intelligent Systems and Technology (TIST), 7(2), 1-22.
>
> [2] Stegle, O., Janzing, D., Zhang, K., Mooij, J. M., & Schölkopf, B. (2010). Probabilistic latent variable models for distinguishing between cause and effect. Advances in neural information processing systems, 23.

---

### Official Review · Reviewer_7wPL · 2022-10-26

**Confidence:** 5
**Correctness:** 3
**Technical Novelty And Significance:** 3
**Empirical Novelty And Significance:** 2
**Recommendation:** 3

**Clarity, Quality, Novelty And Reproducibility:**

The work focuses on an important problem for causal discovery. It is in general well-written. The related points can be found in the previous comments.

**Strength And Weaknesses:**

## Writing
(+) The paper is well-written in general and makes the problem that it is solving and the contributions clear.

(-) However, I would also suggest spending more effort on the proposed method (Page 6 - Page 7) than illustrating the problem (Page 2 - Page 5). Because I would expect the readers can know some about the problems but not really about the proposed method. It can be not the case but then perhaps readers care more about the concerns of the proposed method.

## Experiments
(+) The work provides transparent and nice-illustrated figures.

(-) But in Table 2, it cannot show the proposed method stands out compared with the others empirically. A fairer claim is that it is better than the related works based similar framework. As for real-world data, the ANM-based method performs surprisingly well and is comparable with the proposed method.

(-) Moreover, in this case, it would be good to show experimental results on the typical benchmark dataset, the 100 (or more) Tuebingen cause-effect pairs. Not necessary to show superiority, but to provide a comparison with the others.

## Concerns about Algorithm 2
(-) The causal score is a single number of the difference between the dependency measures in different directions. It would be more convincing to report the numbers for the experiments, especially to which degree (or within which threshold), we can believe the number is larger than zero, smaller than zero, or equal. Moreover, I am worried that in real-world data or finite samples, how much we can trust a single number that gives us the causal relationship. And it doesn't seem to show the experiments about the independent case.


## Proposed method
(+) The work clearly formulates the proposed method with a discussion of optimization process.

(-) Nevertheless, __my main concern__ is that the proposed objective (13) lacks a justification and theoretical guarantee from the identifiability concern. ( Note that I am not asking for an identifiability proof, but the justification and theoretical guarantee for that given the PNLs under identifiability assumptions, can the method be used for determining the causal direction?)

As actionable feedback, can the authors show that
(i) for objective (8), is it in any way related to maximum likelihood or minimizing mutual information?

(ii) as for the model in the causal direction and the one in the reverse causal direction, can objective (8) imply that the one in the causal direction has a smaller value of the optimal objective function value than the other one?

Furthermore,

(iii) as for objective (13), how can the property maintain by adding the penalty term? A concern is that the penalty term is added as a "soft" constraint which is not necessary to be exactly the case. Then, is it possible that the optimal solution by solving (13) can be the local minima which are taken as a trade-off between the objective (8) and the penalty term? Then, will this lead to a misspecified model ? will this lead to a problem for causal discovery?


To further elaborate on my point:
The authors introduced the PNL-MLP of Zhang & Hyvärinen (2009), which uses mutual information for estimating the model and later uses independent tests for causal discovery. And the authors point out the problem of using mutual information as the objective, which can be hard with large-scale datasets. But an important fact of using the maximum likelihood or minimizing mutual information is that they are well justified by the identifiability of PNLs as illustrated in [1], especially, the independent noise assumption.
Similarly, as for the regression by dependence minimization (Mooij et al., 2009), it directly minimizes the HSIC score to enforce the independent noise assumption and pick up the model with a smaller score as the causal direction. This is fine because it directly uses the assumption as objective from the perspective. But for the objective in (Uemura & Shimizu 2020) and this paper, they neither directly use the independent noise assumption nor have a theoretical guarantee of the identifiability as Thm.2 and Thm. 3 in [1]. Therefore, to fix my concerns, maybe the authors could consider my actionable feedback.

[1] Zhang, K., Wang, Z., Zhang, J., & Schölkopf, B. (2015). On estimation of functional causal models: general results and application to the post-nonlinear causal model. ACM Transactions on Intelligent Systems and Technology (TIST), 7(2), 1-22.




**Summary Of The Paper:**

The paper focuses on the challenges of estimating post-nonlinear models (PNLs) for causal discovery in the bivariate case. Indeed, solving the practical issues of estimating PNLs is an important topic.

The problems with existing methods are that they can produce local minima which are trivial and meaningless solutions and that the misspecification of noise distributions can lead to wrong causal discovery results.
The paper proposes an objective that avoids relying on noise distributions (similar property as HSIC minimization methods) and enforces the independence of residual and potential direction cause.

It combines the objective function in the alternating conditional expectation (ACE) algorithm (Breiman & Friedman, 1985) with a dependency measure as the penalty term for enforcing the residual to be independent of the potential direct cause. And it determines the causal direction by using the independent tests of the residual and potential direct cause.

The main experiments in Table 2 cannot show that the proposed method is significantly better than the other methods empirically.


**Summary Of The Review:**

The paper works on the practical issues of estimating PNLs and shows the problems of existing methods. It proposes to use the objective,  maximum correlation with a dependency measure as the penalty term. My main concern is about the theoretical guarantee and the identifiability of the results given by the proposed objective. My minor concern is about the experiments and the causal score used in Algorithm 2.

---

> ### Author Response · Authors · 2022-11-16
> **To Reviewer 7wPL**
>
> Thank you so much for your detailed and constructive review comments.
> Our responses are as follows.
>
> **On Writing:**
>
> Space adjustment. We’ve already saved some spaces in section 2 for the theoretical results of our proposed method (connections to the maximum likelihood).
>
> **On Experiments:**
>
> - **Performance does not stand out:** We admit our method does not stand out all the time, however, it can additionally provide transparent and meaningful feature transformations that most of these SOTA methods can not.
> - **Tuebingen data:** we have noticed this important benchmark dataset early on. In many literatures [1,2], people preferred to use a subset of this dataset for their evaluation. However, we could not find any record on how did they select the subsets to reproduce notable results. So for fairness, we did not report the corresponding results in our paper.  Here, we provide a comparison on Tuebingen data version 1.0. We adopted the implementation of ANM, CDS, IGCI, and RECI in the causal discovery tool package. On this dataset, we can see MC-PNL achieves the same result as AbPNL, with much lower computational complexity.
>
> |     | ANM   | CDS   | IGCI  | RECI  | CDCI  | AbPNL | ACE   | MC-PNL |
> |-----|-------|-------|-------|-------|-------|-------|-------|--------|
> | AUC | 0.552 | 0.667 | 0.657 | 0.705 | 0.722 | 0.567 | 0.537 | 0.567  |
>
> **On Concerns about Algorithm2 (single number causal score):**
> - Reasonable concerns! In the paper, the presentation of causal direction prediction simply follows the bivariate causal discovery framework proposed in [3]. A typical remedy is to use bootstrap, which can give an empirical distribution of the causal score $C$. We have added this comment in the revised manuscript, see supplement I.
> - Experiment-wise, unlike most of the baseline algorithms, ours is insensitive to the initializations (see Figure 3), that is why we did not report any experiments with random seeds.
>
> **On Proposed Method:**
>
> Thanks again for the actionable feedback. Thm 2 and Thm 3 in [1] are good stepping stones to improve our theoretical understanding. We have added the following proposition and the corresponding proof in supplement E.
>
> (i)	**Relation to the maximal likelihood or minimizing mutual information.** (We built the connection for invertible f,g, and Gaussian noise)
>
> **Proposition:** Suppose the dataset $D={(x_i,y_i)}_{i=1}^n$ is generated from a PNL model  $Y = g^{-1}(f(X)+ \epsilon)$, where $f,g$ are both **invertible functions**, and the noise $\epsilon$ follows a Gaussian density $p(\epsilon;\theta)$ with zero mean and variance $\theta$, then maximizing the log-likelihood $\log p(D)$ is equivalent to solving the regression problem (8).
>
> (ii)	**Can objective (8) imply that the one in the causal direction has a smaller value of the optimal objective function value than the other one?**
>
> That is an interesting question to discuss. First, we are not using the scale of residual for causal direction prediction directly. The main purpose of introducing maximal correlation is for better PNL learning with transparent interpretability.
>
> We are not sure if the residual in (8) itself is readily a good indication or not. This can be done in our future work.
>
> (iii)	**On objective (13).**
> With the invertibility and Gaussianity assumptions, there is no trade-off between the two terms. However, in general, the trade-off exists and is controlled by the hyper-parameter $\lambda$. Nevertheless, objective (13) can still guide us to a relatively meaningful solution. Based on this, a fine-tuned process by enlarging the $\lambda$ to 10000 was introduced in supplement G (H in the updated version) to further correct the model. This is much better than the original PNL-MLP with blind or human-involved initialization.
>
> We have to emphasize that the issues of previous PNL-based methods are met when dealing with a small dataset (in terms of sample size). When the model expressiveness is high (e.g., deep neural networks), with finite samples, the previous methods (minimizing MI/HSIC only) are not practical to use because they can fit any independent noise. Our objective (13), somehow, is analogous to the L2+HSIC loss, and the Soft-HGR term is to learn meaningful representations, while the HSIC term targets independent residuals.
>
> [1] Zhang, K., Wang, Z., Zhang, J., & Schölkopf, B. (2015). On estimation of functional causal models: general results and application to the post-nonlinear causal model. ACM Transactions on Intelligent Systems and Technology (TIST), 7(2), 1-22.
>
> [2] Stegle, O., Janzing, D., Zhang, K., Mooij, J. M., & Schölkopf, B. (2010). Probabilistic latent variable models for distinguishing between cause and effect. Advances in neural information processing systems, 23.
>
> [3] Mooij, J. M., Peters, J., Janzing, D., Zscheischler, J., & Schölkopf, B. (2016). Distinguishing cause from effect using observational data: methods and benchmarks. The Journal of Machine Learning Research, 17(1), 1103-1204.

---

### Decision · Program_Chairs · 2023-01-20

**Decision:**

Reject

**Justification For Why Not Higher Score:**

Given that causal discovery aims to find true causal direction, the paper will benefit a lot from suitable theoretical guarantees.

**Justification For Why Not Lower Score:**

The paper is in general well written, and the tackled problem is interesting and has direct practical implications.

**Metareview: Summary, Strengths And Weaknesses:**

The paper focuses on the challenges of estimating post-nonlinear models (PNLs) for causal discovery in the bivariate case. The proposed method combines the objective function in the alternating conditional expectation (ACE) algorithm with a dependency measure as the penalty term for enforcing the residual to be independent of the potential direct cause. It then determines the causal direction by using the independent tests of the residual and hypothetical direct cause. The paper is in general well written, and the tackled problem is interesting and has direct practical implications. However, given that causal discovery aims to find true causal direction, the paper will benefit a lot from suitable theoretical guarantees.